# Affinity and cooperativity modulate ternary complex formation to drive targeted protein degradation

Ryan P. Wurz[1], Huan Rui[1], Ken Dellamaggiore[1], Sudipa Ghimire-Rijal[1], Kaylee Choi[2], Kate Smither[1], Albert Amegadzie[1], Ning Chen[1], Xiaofen Li[1], Abhisek Banerjee[3], Qing Chen[1], Dane Mohl[1] ✉ & Amit Vaish[1] ✉

Targeted protein degradation via "hijacking" of the ubiquitin-proteasome system using proteolysis targeting chimeras (PROTACs) has evolved into a novel therapeutic modality. The design of PROTACs is challenging; multiple steps involved in PROTAC-induced degradation make it difficult to establish coherent structure-activity relationships. Herein, we characterize PROTAC-mediated ternary complex formation and degradation by employing von Hippel–Lindau protein (VHL) recruiting PROTACs for two different target proteins, SMARCA2 and BRD4. Ternary-complex attributes and degradation activity parameters are evaluated by varying components of the PROTAC's architecture. Ternary complex binding affinity and cooperativity correlates well with degradation potency and initial rates of degradation. Additionally, we develop a ternary-complex structure modeling workflow to calculate the total buried surface area at the interface, which is in agreement with the measured ternary complex binding affinity. Our findings establish a predictive framework to guide the design of potent degraders.

Several therapeutic modalities have emerged wherein ternary complex formation is critical to their mechanism of action. These include bispecific recombinant proteins and antibody agents (e.g., BiTEs and DART platforms) that activate innate T-cells to direct their activity toward tumor cells, and molecular glues (i.e., IMiDs, aryl-sulfonamides) that direct the activity of a protein complex toward a neo-substrate[1–3]. In particular, molecular glues drive ternary complex formation by providing necessary protein-glue-protein contacts and by promoting new protein-protein contacts[4]. A well-studied example of a naturally occurring molecular glue is auxin (indole-3-acetic acid or IAA), a plant hormone that regulates growth and development by orchestrating the degradation of a family of transcription factors through a conserved degron[5]. When auxin is present, ubiquitin ligase SCF[Tir1] gains the ability to bind auxin inducible degron (AID) and promote ubiquitination and eventual destruction of the transcription factor by the ubiquitin proteasome system (UPS)[6]. Crystallographic studies of SCF[Tir1]-IAA-AID

show auxin occupying a pocket located within the TIR1 substrate recognition domain, patching a hole in the degron binding site and completing a hydrophobic surface that drives protein-AID association[7,8]. In contrast to IAA, the natural product rapamycin induces a FKB12-rapamycin-FRB ternary complex through protein-drug-protein interactions over a large surface provided by the small-molecule[9]. Similar to auxin, the consequences of rapamycin induced proximity are dramatic; treatment with rapamycin produces strong pharmacological effects that include anti-fungal, immunosuppressant, and anti-cancer activities[10].

Inspired by auxin and rapamycin, researchers have sought to engineer small molecules that induce protein proximity in order to prospectively target proteins of pharmacological interest and to develop chemical genetic tools[11]. Proteolysis targeting chimeras (PROTACs) are heterobifunctional molecules engineered by linking two small molecules, one that binds an E3 ubiquitin ligase and a second

[1]Amgen Research, Amgen Inc., Thousand Oaks, CA, USA. [2]Amgen Research, Amgen Inc., South San Francisco, CA, USA. [3]Syngene Amgen Research Center (SARC), Bangalore, India. ✉e-mail: dmohl@amgen.com; avaish@amgen.com

ligand that recognizes a target protein[12–16]. Inducing proximity between a target protein and a ubiquitin ligase complex can promote the transfer of ubiquitin to the target and subsequent degradation through the 26 S proteasome[12,17]. PROTACs need not functionally inhibit or perturb an active site, thus this emerging modality represents a promising strategy for therapeutic intervention for diseases that are driven by traditionally "difficult-to-drug" proteins[1,18,19].

PROTACs are constructed from two ligands that are connected by a linker. The nature of this linker, length and composition, and the attachment site of the linker to each ligand affects the affinity of ligands to each of its binding partners and the relative orientation of the two proteins to one another in the ternary complex. *In lieu* of a rational approach for designing linkers[20], analogs with differing lengths, compositions, and linkage vectors can be systematically evaluated to tune the PROTAC[21–24]. Linker and ligand structure-activity relationships (SAR) have proven difficult to predict as degradation requires both ternary complex formation and ubiquitination. For example, a high affinity ternary complex may not necessarily facilitate ubiquitination of the target protein if lysine residues on the surface of the substrate protein are inaccessible to ubiquitin loaded E2 proteins (Fig. 1a)[25]. The composition of the linker also impacts the physiochemical properties of the PROTAC and needs to be tuned to achieve desired solubility, permeability, and bioavailability. To assist these efforts, the research community employs a host of cell-based, biochemical, and biophysical assays.

PROTAC-mediated ternary complex formation has been analyzed by proximity-based assays in which a bifunctional molecule is titrated against two labeled proteins to generate a dose-response (DR) curve[24,26]. The salient characteristic of these binding interactions is a bell-shaped curve that describes the formation of a ternary complex and its dissolution due to competing binary interactions. Although these assays provide a means to evaluate the potency of PROTACs relative to one another, additional methods are required to

characterize the thermodynamic and kinetic parameters of the complex[27]. Ciulli et al. described a methodology to directly measure the binding affinity of PROTAC-mediated ternary complex formation using label-free techniques such as surface plasmon resonance (SPR) or isothermal titration calorimetry (ITC)[26,28]. These methods avoid measuring the competing binary interactions by using a preformed binary complex consisting of PROTAC and excess target protein (Fig. 1c). SPR and ITC methods reveal that ternary complex formation can be potently influenced by both protein-protein interactions (PPIs) and protein-small molecule interactions. This phenomenon is defined as cooperativity ($\alpha$) and is derived by calculating the ratio of a PROTAC's binary to ternary complex binding affinity[24,26,28–30]. Positive cooperativity ($\alpha > 1$) results from higher ternary complex binding affinity compared to corresponding binary complex due to favorable PPIs at the interface. Conversely, negative cooperativity ($\alpha < 1$) is a potential consequence of unfavorable PPIs.

To guide our own efforts to engineer potent and efficient PROTACs, we sought to describe the relationship between the initial step (ternary-complex formation) and the final step (target degradation) of the PROTAC-mediated degradation pathway (Fig. 1a). In this work, we measure ternary complex formation using SPR and determine cellular potency and degradation rates for two series of PROTACs (Fig. 1b, c). We chose to target both SMARCA2 and BRD4 because they encode druggable bromodomains for which ligands have already been identified. For each series, SMARCA2 or BRD4 targeting ligands are linked to a previously reported high affinity VHL binder[22,26,31,32] (Tables 1 and 2). We also tested MZ1[26], a BRD4 PROTAC that is well characterized and is understood to induce ternary complex formation through cooperative protein-protein interactions. Using these molecules, the relationships between ternary complex binding parameters (i.e., affinity, cooperativity) and PROTAC activity in cells are evaluated and a predictive framework for advancing novel and efficient degraders is established. We show that simulations of PROTAC-induced ternary complexes enable

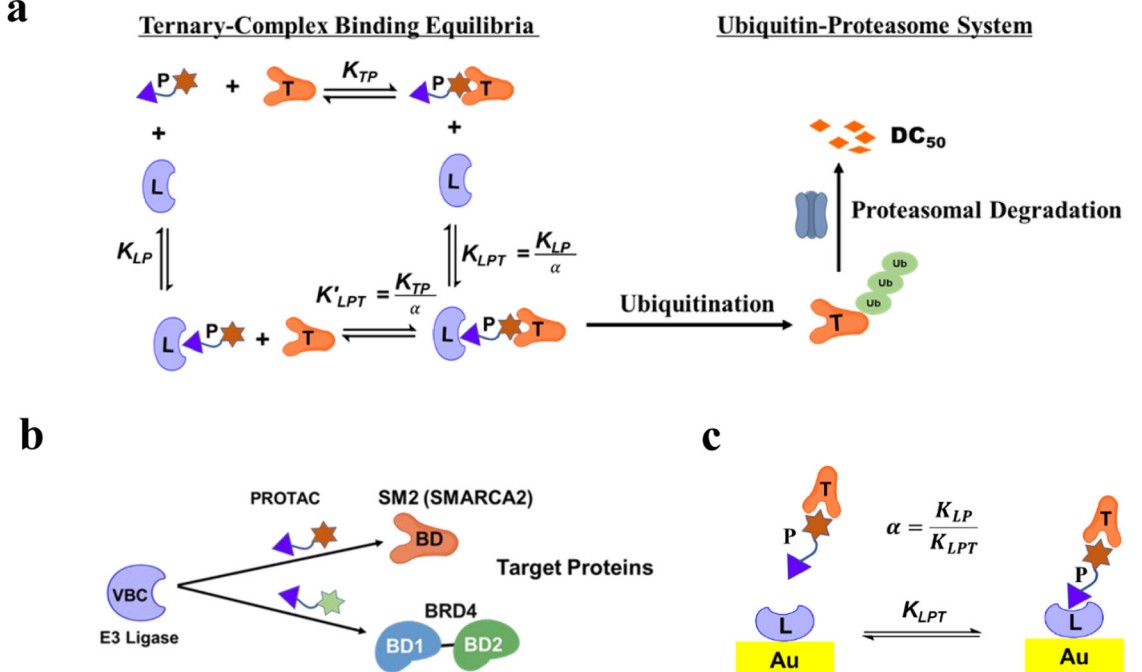

**Fig. 1 | Characterization of degrader-induced ternary complex formation.**
**a** Schematic illustration of the thermodynamic cycle of heterobifunctional molecule (*P*) induced ternary complex formation between an ubiquitin ligase (*L*) and a target protein (*T*). The equilibrium between *L*, *P* and *T* can be divided into four probable equilibria involved in ternary complex formation: the binary equilibrium between *P* and *L* or *T* is characterized by binding affinity $K_{TP}$ or $K_{LP}$, and subsequent binding

equilibrium between binary complex (*LP* or *TP*) and *T* or *L* is characterized by a ternary complex binding affinity $K_{LPT} = K_{LP}/\alpha$, where $\alpha$ is the cooperativity factor. A target protein in the ternary complex can be ubiquitinated by the E3 ligase machinery, followed by proteasomal degradation. **b** VHL/elongin B/elongin C (VBC) engagement with SMARCA2 or BRD4 PROTACs. **c** SPR assay involving VBC-functionalized surface to evaluate binding affinity of PROTAC-mediated ternary complexes [*LPT*].

## Table 1 | SMARCA2-VHL PROTACs[a]

| Cmpd | R | X | Linker | DC$_{50}$ (nM) [Dmax(%)] |
|---|---|---|---|---|
| 1 | NH$_2$ | CH | (2,4-pyridinyl) | 28 (98) |
| 2 | NH$_2$ | CH | (2,6-pyridinyl) | 95 (95) |
| 3 | NH$_2$ | CH | (2,5-pyridinyl) | >10000 |
| 4 | NH$_2$ | CH | (1,3-phenyl) | 99 (96) |
| 5 | NH$_2$ | CH | (2,5-thiazolyl) | 112 (96) |
| 6 (AU-15330) | NH$_2$ | CH | (ethylene) | 85 (96) |
| 7 | NH$_2$ | CH | (pyridinyl-CH$_2$) | 850 (55) |
| 8 | NH$_2$ | CH | (pyridinyl-CH$_2$CH$_2$) | 379 (80) |
| 9 | NH$_2$ | N | (2,4-pyridinyl) | 372 (74) |
| 10 | H | CH | (2,6-pyridinyl) | 469 (65) |

## Table 1 (continued) | SMARCA2-VHL PROTACs[a]

| Cmpd | R | X | Linker | DC$_{50}$ (nM) [Dmax(%)] |
|---|---|---|---|---|
| 11 | | | | 221 (53) |

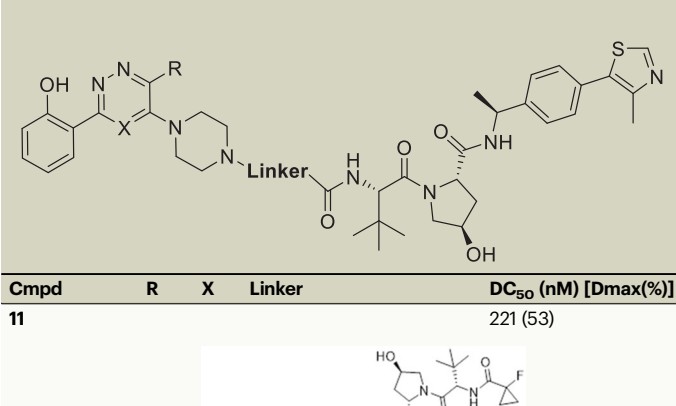

[a]SMARCA2 degradation as measured by MSD assay in A375 cells (2 h timepoint).

the calculation of the total buried surface area (BSA) at the interface which correlates with measured ternary complex binding affinity. This suggests that prospective engineering of a PROTAC can be guided in part by in silico experimentation.

## Results

### Mathematical framework for describing SPR-derived binding parameters of PROTAC-mediated ternary complex formation

To better understand the relationship between the biophysical parameters that define PROTAC-mediated ternary complex formation (i.e., binding affinity, cooperativity) and those that describe cellular activity (i.e., potency, degradation rate), we sought to analyze the mathematical framework involved in these intricate processes. In the SPR assay (Fig. 1c), the equilibrium dissociation constant ($K_{LPT}$) for the interaction between a surface-bound ligase ($L$) and a preformed binary complex ($TP$) involving a target ($T$), a corresponding PROTAC ($P$) and total bound ligase [$LPT$] at equilibrium is defined as:

$$LPT \rightleftharpoons L + TP \tag{1}$$

$$K_{LPT} = \frac{[L][TP]}{[LPT]} \tag{2}$$

As discussed in Appendix, eq. 2 can be further modified and written as:

$$\frac{[LPT]}{[L]_t} = \frac{[P]_t}{[P]_t + K_{LPT}} \tag{3}$$

Where, [$L$]$_t$ is total surface-immobilized ligase, and [$P$]$_t$ is total PROTAC concentration. Equation 3 can be fitted to an asymptotic/sigmoidal curve of varying concentrations of PROTAC [$P$]$_t$ involved in forming ternary complex [$LPT$] on a surface-immobilized ligase [$L$]$_t$, and $K_{LPT}$ will be equal to the PROTAC concentration [$P$]$_t$ when half of the [$L$]$_t$ is engaged in forming the ternary complex (Fig. 2a).

The previous work of Douglass et al.[33] on three-body equilibrium binding (Fig. 1a) was adapted to analyze the PROTAC-mediated complex formation by SPR (Fig. 1c). The mathematical equation describing ternary complex equilibrium of a cooperative system is algebraically

## Table 2 | BRD4-VHL PROTACs[b]

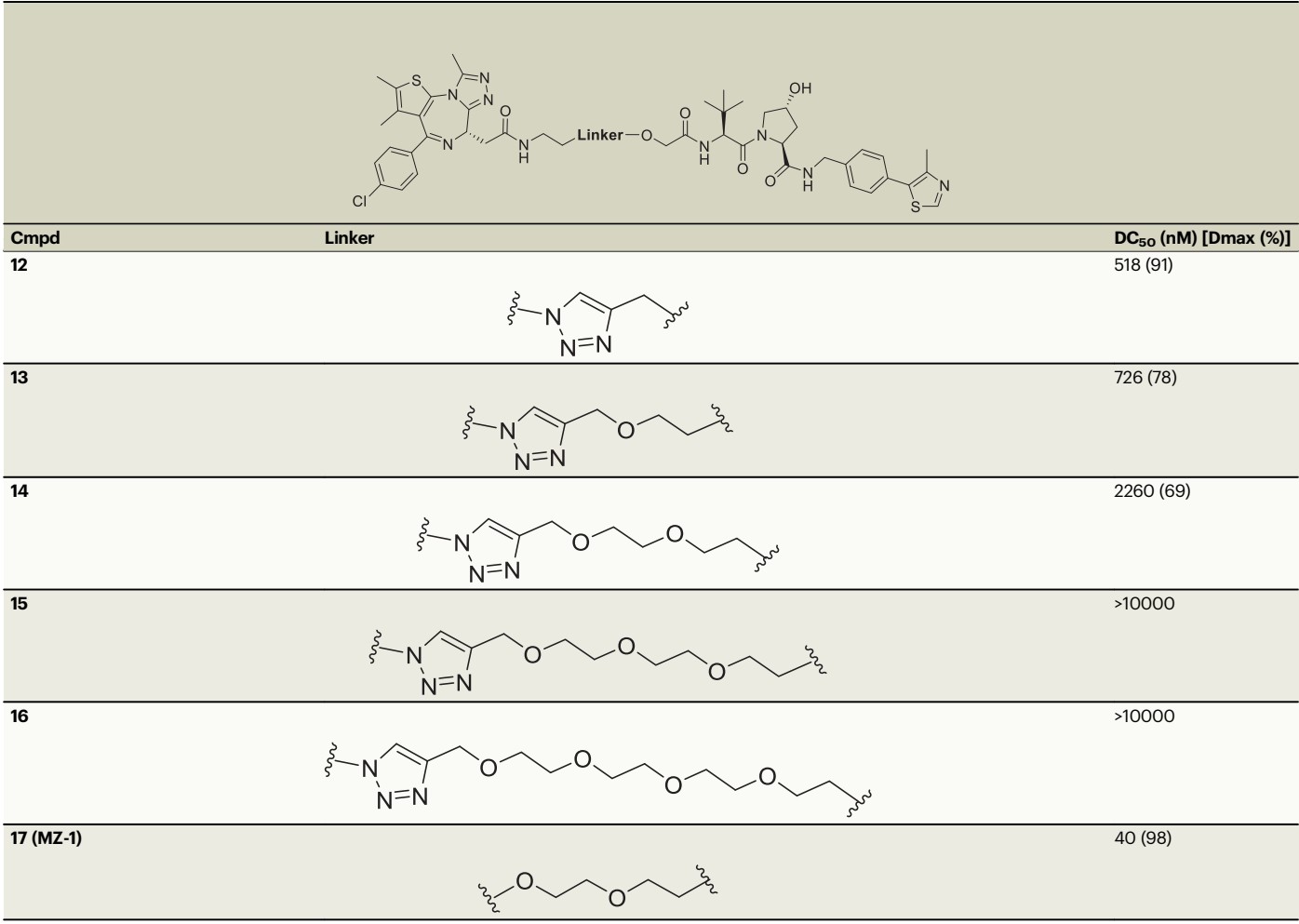

| Cmpd | Linker | DC$_{50}$ (nM) [Dmax (%)] |
|---|---|---|
| 12 | | 518 (91) |
| 13 | | 726 (78) |
| 14 | | 2260 (69) |
| 15 | | >10000 |
| 16 | | >10000 |
| 17 (MZ-1) | | 40 (98) |

[b]BRD4 degradation as measured by MSD assay in A375 cells (4 h timepoint).

unsolvable, however, it can be solved for maximal ternary complex. In the SPR experimental setup (Fig. 1c), a reaction mixture comprising of binary complex (TP) and excess target protein (T) is introduced to a ligase-functionalized surface inside the SPR flow cell. In this laminar flow configuration (Fig. 2b), the reactant mass flux is balanced by binding interactions on the surface. The mathematical equation derived for maximal ternary complex is modified by incorporating the SPR constraints (vide Supplementary Information) as follows:

$$\frac{[LPT]_{\max}}{[L]_t} \cong \frac{\alpha}{\left(\alpha + \frac{(\sqrt{K_{LP}} + \sqrt{K_{TP}})^2}{[T]_t}\right)} \qquad (4)$$

Where maximal ternary complex formation, $[LPT]_{\max}$, is represented by the SPR response at top $[P]_t$, and cooperativity ($\alpha$) is the ratio of binary binding affinity (ligase/PROTAC; $K_{LP}$) to ternary binding affinity (ligase/PROTAC/target; $K_{LPT}$). In the SPR assay, $[T]_t \approx 25 \times K_{TP}$ for maintaining the binary complex (vide Supplementary Information). Therefore, Eq. 4 can be rewritten as:

$$\frac{[LPT]_{\max}}{[L]_t} \cong \frac{\alpha}{\alpha + \frac{\left(\sqrt{\frac{K_{LP}}{K_{TP}}} + 1\right)^2}{25}} \cong \frac{\alpha}{\alpha + \beta} \qquad (5)$$

Here $\beta = \frac{\left(\sqrt{\frac{K_{LP}}{K_{TP}}} + 1\right)^2}{25}$, which accounts for the corresponding binary binding affinities of the PROTAC to ligase and target. Figure 2c illustrates curves of ternary complex fraction ($[LPT]_{\max}/[L]_t$) versus cooperativity ($\alpha$) as a function of $\beta$. As shown in Fig. 2c, $[LPT]_{\max}/[L]_t$ increases with $\alpha$ for different values of $\beta$. For $\beta = 0.2$ (when $K_{LP} = K_{TP}$), $[LPT]_{\max}/[L]_t$ reaches 0.9 at $\alpha = 2$, beyond which it doesn't change much with the increase of $\alpha$. In contrast, higher cooperativity is required for systems with higher $\beta$ (> 0.5) to achieve the ternary complex fraction of 0.9. Therefore, the extent of maximal ternary complex formation, $[LPT]_{\max}$, for a given system could be modulated by the $\alpha$ factor of a PROTAC (Fig. 2a).

The target protein drawn into a PROTAC-induced ternary complex can be ubiquitinated by ubiquitin-conjugating enzyme (E2) as shown in Fig. 1a. By combining E2 enzyme activity to E3 ligase complex, an expression analogous to Segel[34] and Vieux et al.[35], under the assumption of rapid equilibrium[36] can be written for target ubiquitination initial rate as follows:

$$\upsilon = \frac{V_{\max}}{(1 - \frac{K_{LPT}}{K_{LP}}) + (1 + \frac{[P]_t}{K_{LP}}) \frac{2K_{LPT}}{([P]_t + [T]_t + K_{TP}) - \sqrt{([P]_t + [T]_t + K_{TP})^2 - 4[P]_t[T]_t}}} \qquad (6)$$

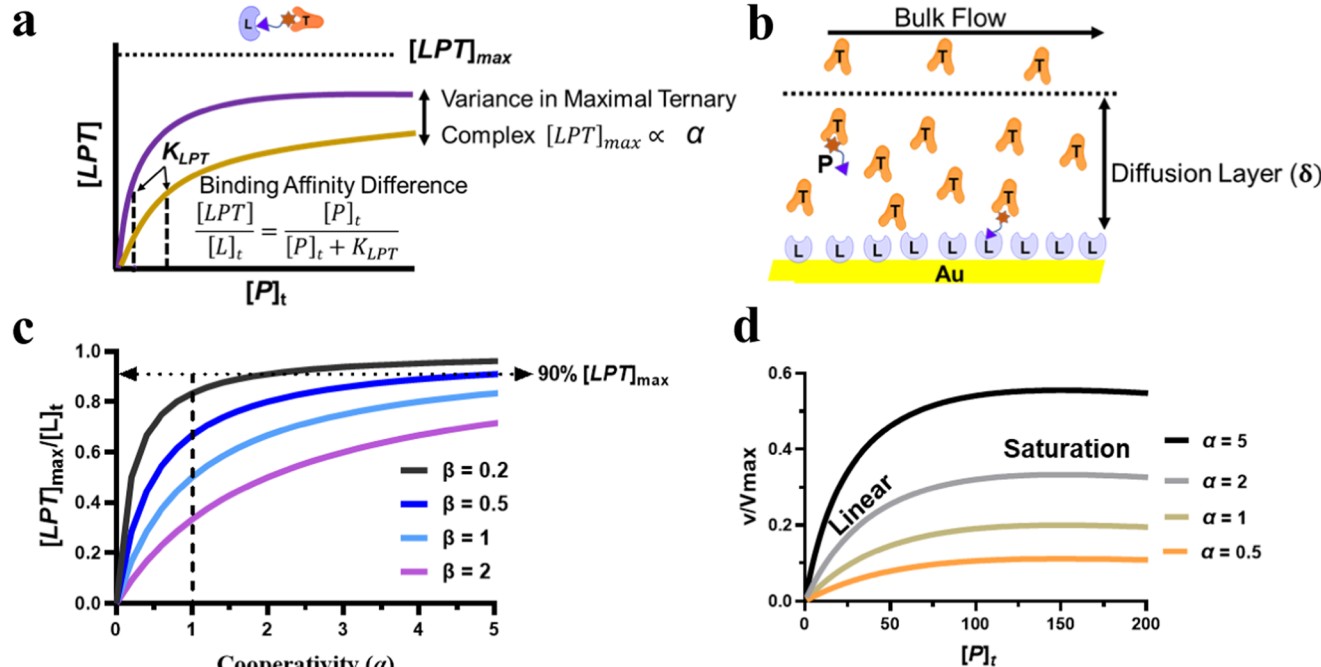

**Fig. 2 | Binding parameters influence ternary complex formation. a** Illustration of the differences in binding affinities (Eq. 3) and cooperativity (Eq. 4) between two PROTACs. **b** Schematic of SPR flow cell with a target protein ($T$) and a PROTAC ($P$) on a ligase-functionalized ($L$) chip, where the thickness of the diffusion layer is $\delta$. **c** Plots of PROTAC-mediated ternary complex fraction ($[LPT]_{max}/[L]_t$) on a SPR chip as a function of cooperativity ($\alpha$) with varying $\beta$. The dotted arrow indicates the 90% $[LPT]_{max}$ onto a SPR chip. **d** Plots representing initial target ubiquitination rate versus total PROTAC concentration $[P]_t$ by varying $\alpha$ in the presence of fixed $[L]_t$, $K_{LP}$, and $K_{TP}$.

Modifying Eq. 6 to replace ternary complex affinity with cooperativity factor ($\alpha = K_{LP}/K_{LPT}$),

$$\frac{\nu}{V_{max}} = \frac{\alpha}{\alpha + \left(\frac{2(K_{LP}+[P]_t)}{([P]_t+[T]_t+K_{Tp})-\sqrt{([P]_t+[T]_t+K_{TP})^2-4[P]_t[T]_t}} - 1\right)} \quad (7)$$

where $\nu$ is target ubiquitination initial rate, and $V_{max}$ is the product of ternary complex breakdown rate constant and total ligase concentration. Figure 2d shows a series of plots of normalized target ubiquitination initial rate ($\nu/V_{max}$) as a function of PROTAC concentration $[P]_t$ under different cooperativity ($\alpha$) in the presence of a fixed value of 100 nM for the three parameters: $[T]_t$, $K_{LP}$, and $K_{TP}$. As shown in Fig. 2d, all of the curves have two characteristic regions: (1) a linear region at lower $[P]_t$ and (2) a saturation region at a very high $[P]_t$.

Linear regime: Eq. 7 can be further modified by assuming $[P]_t \ll [T]_t$ or $K_{LP}$ or $K_{TP}$, and ignoring the lower-order terms, and subsequently replacing cooperativity ($\alpha = K_{LP}/K_{LPT}$) with ternary complex binding affinity as follows:

$$\frac{\nu}{V_{max}} \approx \frac{[P]_t}{2K_{LPT}} \quad (8)$$

Equation 8 suggests that the initial ubiquitination rate is dependent on the ternary complex binding affinity and PROTAC concentration in the linear regime. As discussed earlier, PROTAC-mediated ternary complex formation is required for target degradation via UPS[11,36,37] and binding affinity measures the strength of this tripartite binding interaction (Fig. 2a). Consequently, a degradation activity parameter derived from a DR curve (i.e., DC$_{50}$)[38] could be correlated with the ternary complex binding affinity.

Saturation regime: similarly, Eq. 7 can be further modified by assuming $[P]_t \gg [T]_t$ or $K_{LP}$ or $K_{TP}$, and ignoring the lower-order terms

as follows:

$$\frac{\nu}{V_{max}} \approx \frac{\alpha[T]_t}{2[P]_t} \quad (9)$$

Here, Eq. 9 suggests that the target ubiquitination rate is directly correlated to the cooperativity factor of PROTAC at a saturation concentration. The initial degradation rate reflects the efficiency of a PROTAC in degrading a target at a concentration with maximal degradation activity. As shown in Fig. 1a, ubiquitinated target can be degraded by the proteasome, and the initial rate of degradation can be assumed to be proportional to the target ubiquitination initial rate[37,39]. Therefore, initial rate of target degradation could be correlated to the PROTAC's cooperativity factor. As illustrated in Fig. 2c, cooperativity is the key factor driving maximum PROTAC-mediated ternary complex formation.

### Binding affinity influences degradation potency of VHL-based PROTACs

To probe the relationship between ternary-complex formation and induced protein degradation, two series of VHL dependent PROTACs were generated, one targeting SMARCA2 (Table 1) for degradation, and one to promote BRD4 (Table 2) degradation[31,40]. Each of these series were characterized by SPR to determine ternary complex binding affinity, stability, and cooperativity while dose response and single dose time-course assays were used to assess cellular activity (Figs. 4, 5). DC$_{50}$, the concentration at which 50% of the target is degraded, has been adopted by the PROTAC community as a descriptor of potency[22,38]. For this work, we calculated both DC$_{50}$ and area under the curve (AUC) to describe degradation activity. AUC is particularly useful for PROTACs where <50% of protein is degraded relative to the control treatment and a DC$_{50}$ cannot be calculated. To explore the relationships between measured SPR parameters and cellular activity, we plotted DC$_{50}$, AUC, and initial rate of degradation

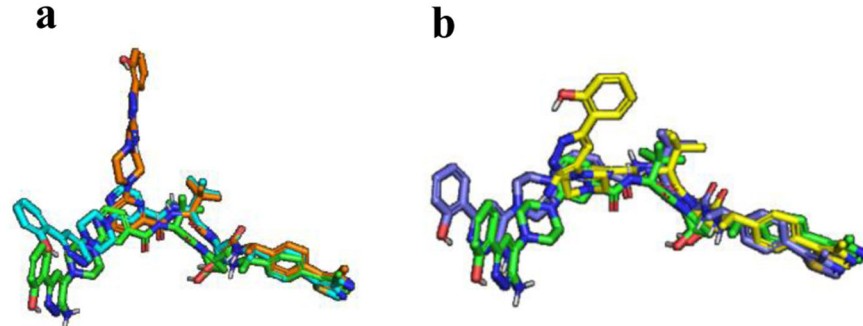

**Fig. 3 | Model structures of PROTACs. (a) 1-2-3**, and **(b) 1-4-5** from the best structural model of the ternary complexes. The PROTACs **1** (green), **2** (orange), **3** (cyan), **4** (yellow), and **5** (slate). All PROTACs are in the same frame by aligning VHL.

against ternary complex affinity, cooperativity, and half-life (Figs. 4, 5, Supplementary Fig. 2). These scatterplots enabled us to identify SAR in our molecules that exemplified the relationships between ternary-complex formation and cellular activity which are identified by the mathematical framework outlined in this manuscript (vide supra).

Orientation of the ligase with respect to the target plays a key role in PROTAC-mediated ternary-complex formation[22]. Our linker optimization efforts revealed that PROTACs bearing minimalistic linkers can induce favorable protein-protein interactions between SMARCA2 and the E3 ubiquitin ligase when the appropriate linker vector is identified. PROTACs (**1-3**) bearing a common pyridine linker differ in the way the ligands (SMARCA2 and VHL) are positioned with respect to the nitrogen of the pyridine ring (Fig. 3a). A drastic difference in the ternary complex binding affinity ($K_{LPT}$) for these regioisomers was observed with compound **1** promoting the highest affinity complex. In contrast, PROTACs **2** and **3** exhibit one order and two orders of magnitude weaker binding affinity, compared to **1** (Table 3). PROTAC **1** demonstrated robust SMARCA2 degradation within 2 h in the MSD assay (lowest AUC) while **2** led to a right shifted AUC (two times >**1**) and **3** exhibited no degradation (Fig. 4a). Additionally, PROTACs designed with linkers having different functional groups (Fig. 3b) such as a benzene ring (**4**) or a thiazole (**5**) demonstrated one order of magnitude weaker binding affinity than the optimal pyridine linker (**1**). Both **4** and **5** have AUC values of ~26, which is 2-fold >**1**. A literature benchmark, **6** (AU-15330)[32] demonstrated a ternary complex binary affinity similar to **1**, however, as shown in Fig. 4a and reported in Table 3, cellular SMARCA2 degradation activity (AUC) of **6** (AU-15330) is ~2-fold <**1**, which could be attributed to its poor cellular permeability compared to **1** (Supplementary Table 8). Changing the linker length, which is an important PROTAC design parameter, revealed that elongation of the linker connecting the two ligands (**7** and **8**) resulted in a drop in ternary complex binding affinities by one order of magnitude compared to **1**, coupled with inefficient degradation of SMARCA2.

Modification of the SMARCA2 ligand significantly impacted the PROTAC's ability to form a ternary complex and degrade SMARCA2. In reference to **1**, ternary complex binding affinity dropped by an order of magnitude upon incorporation of a SMARCA2 ligand bearing a 1,2,4-triazine motif (**9**) or two orders of magnitude upon removal of the amine from the 6-aminopyridazine scaffold (**10**). Additionally, **9** and **10** were much less efficient SMARCA2 degraders (~30% degradation), compared to PROTAC **1** in the 2 h MSD assay (Fig. 4a). Another literature benchmark, **11** (PROTAC 2)[31], demonstrated weak ternary complex binding affinity as well as less efficient degradation of SMARCA2. To assess the impact of ternary complex formation on PROTAC induced degradation, ternary complex binding affinity was plotted against AUC and $DC_{50}$ (Fig. 4c, d). For the SMARCA2 series of VHL-dependent PROTACs, binding affinity ($K_{LPT}$) demonstrated strong positive correlation with both AUC ($r = 0.79$), and $DC_{50}$ ($r = 0.76$). Conversely, AUC demonstrated negative correlation ($r = -0.76$) with cooperativity (Supplementary Fig. 1).

VHL-recruiting BRD4 PROTACs were studied to discern the commonalities between protein degradation via the E3 ubiquitin ligase VHL. A small library of analogs with varying linker lengths[38] (Table 2) were tested along with the literature benchmark MZ1 (**17**). PROTAC dose response curves in the MSD assay (Fig. 5a) suggested that BRD4 PROTACs degradation activity diminishes with increasing linker length as compared to **17** (MZ1). We sought to analyze ternary complex formation with both bromodomains, BRD4[BD1] and BRD4[BD2], independently to identify the dominant PROTAC-mediated interaction[26,28]. **17** (MZ1) exhibited a faster ternary complex dissociation with BRD4[BD1] compared to BRD4[BD2] (Supplementary Fig. 6), and ternary complex

### Table 3 | Binding and degradation parameters[c]

| Target | PROTAC | $K_{LP}$ (nM) | $K_{LPT}$ (nM) | $\alpha$ | AUC | Initial Degradation Rate (%/min) |
|---|---|---|---|---|---|---|
| SMARCA2 (BD) | **1** | 60 ± 3 | 4.7 ± 1 | 12.8 | 13 | 1.9 |
| | **2** | 210 ± 18 | 33 ± 4 | 6.5 | 28 | 2.02 |
| | **3** | 58 ± 4 | ~500 | | | 0.1 |
| | **4** | 166 ± 15 | 64 ± 7 | 2.6 | 26 | 2.06 |
| | **5** | 84.7 ± 5 | 46.8 ± 6.7 | 1.8 | 28 | 1.9 |
| | **6** (AU-15330) | 11 ± 1 | 5.6 ± 2 | 2 | 26 | 3.5 |
| | **7** | 9 ± 2 | 80 ± 6 | 0.11 | 122 | 0.78 |
| | **8** | 7 ± 2 | 27 ± 3 | 0.26 | 75 | 1.34 |
| | **9** | 25 ± 4 | 77 ± 8 | 0.32 | 85 | 0.98 |
| | **10** | 60 ± 5 | 250 ± 30 | 0.24 | 100 | 0.56 |
| | **11** | 59 ± 5 | 108 ± 14 | 0.55 | 119 | 0.25 |
| BRD4 (BD1) | **12** | 22 ± 2 | 31 ± 4 | 0.7 | 192 | 1.38 |
| | **13** | 17 ± 2 | 29 ± 3 | 0.6 | 309 | 0.76 |
| | **14** | 26 ± 5 | 33 ± 4 | 0.8 | 460 | 0.46 |
| | **15** | 31 ± 4 | 26.5 ± 4 | 1.2 | 733 | 0.21 |
| | **16** | 33 ± 5 | 33 ± 5 | 1 | 900 | 0.08 |
| | **17** (MZ1) | 20.5 ± 3 | 15 ± 2 | 1.4 | 39 | 1.95 |
| BRD4 (BD2) | **12** | 22 ± 2 | 6 ± 1 | 3.7 | 192 | 1.38 |
| | **13** | 17 ± 2 | 10 ± 2 | 1.7 | 309 | 0.76 |
| | **14** | 26 ± 5 | 26.5 ± 4 | 1 | 460 | 0.46 |
| | **15** | 31 ± 4 | 39 ± 5 | 0.8 | 733 | 0.21 |
| | **16** | 33 ± 5 | 49 ± 6 | 0.7 | 900 | 0.08 |
| | **17** (MZ1) | 20.5 ± 3 | 1.5 ± 0.5 | 13.7 | 39 | 1.95 |

[c]$K_{LP}$ represents binary binding affinity between E3 ligase VHL and PROTAC. $K_{LPT}$ represents ternary complex binding affinity between E3 ligase VHL, PROTAC, and SMARCA2 or BRD4. Error is SEMs for $N = 3$.

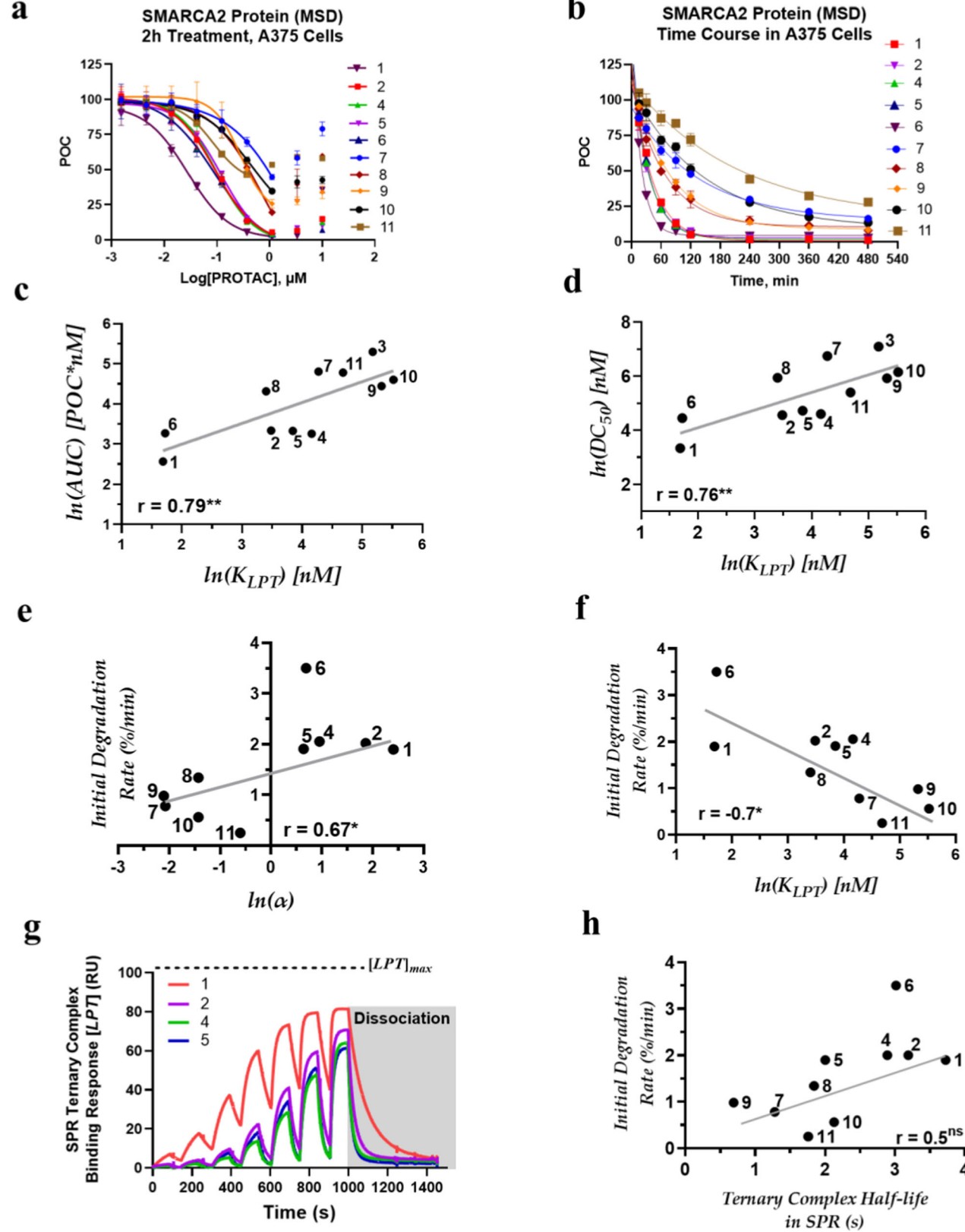

binding affinity (Table 3) with $BRD4^{BD1}$ is 10-fold weaker compared to $BRD4^{BD2}$[26,28,41]. Both measures of degradation efficiency, AUC and $DC_{50}$, demonstrated strong positive correlations ($r = 0.98$) with the ternary complex binding affinity (Fig. 5c, d). Additionally, AUC has shown negative correlation ($r = -0.91$) with the cooperativity (Supplementary Fig. 2).

To build upon these SPR observations, a nanoBRET target engagement assay (Fig. 6), similar to that published by Riching et al., was used to probe intracellular PROTAC-induced ternary complex formation[37]. The ternary complex binding affinity appears to be weaker in cells compared to that for purified proteins, but these two binding affinity parameters correlate well with one another ($R^2 = 0.72$), and the

**Fig. 4 | Relationship between ternary-complex attributes and degradation parameters of SMARCA2 degraders.** SMARCA2 MSD assay data for **a** 2 h time point **b** degradation time course in A375 cells. Error represents SEMs for 3 biologically independent experiments in 2 h time point MSD assay. Time course curves are a best fit (one-phase decay) of means from 3 biologically independent experiments, error represents SEMs. Correlation between cellular and biophysical SAR of SMARCA2 PROTACs. **c** Area under the dose–response curve (AUC) and **d** half-maximal degradation concentration ($DC_{50}$) is plotted against ternary complex binding affinity ($K_{LPT}$) for SMARCA2. Both AUC and $DC_{50}$, show strong correlations with $K_{LPT}$ with **$P = 0.005$ and **$P = 0.007$, respectively. Initial target degradation rate is plotted against **e** cooperativity factor ($\alpha$) and **f** binding affinity ($K_{LPT}$). A positive correlation (*$P = 0.05$) exists between $\alpha$ and initial SMARCA2 degradation rates. Conversely, $K_{LPT}$ show a negative correlation (*$P = 0.01$) with initial SMARCA2 degradation rates. **g** Representative SPR sensorgrams of PROTAC-mediated ternary complex formation between VBC and $SMARCA2^{BD}$ with a gray box depicting dissociation of ternary complex, and **h** scatterplot illustrating weak correlation (ns$P = 0.07$) between initial SMARCA2 degradation rate and half-life (*log*) of SMARCA2 PROTACs in SPR. Pearson correlation coefficient (r) is used for correlation analysis and two-tailed test for significance; nonsignificant (ns) indicates $P > 0.05$.

cellular affinities also correlate well with the AUC of BRD4 degradation. The apparent differences in affinities measured in cells and by SPR could be due to limited cellular permeability of our BRD4 PROTACs (Supplementary Table 9).

### Positive cooperativity promotes higher rates of degradation

Measurement of $K_{LPT}$ for both VHL-SMARCA2 and VHL-BRD4 degraders demonstrated that ternary complex affinity drives potency. This simple relationship allowed us to understand changes in linker length and ligand orientation to arrive at our most potent SMARCA2 degraders. Previous work has demonstrated that PROTAC-mediated ternary complex binding affinity can be higher than that predicted by a PROTAC's binary binding affinity to either ligase or target alone[26]. This characteristic has been described as cooperativity ($\alpha > 1$) and likely results from favorable PPIs present in the ternary complex. During our search for the most potent SMARCA2 degraders, SAR was exquisitely sensitive to orientation of target ligand to ligase ligand (Fig. 3a, b) and this observation convinced us that PPIs, and thus cooperativity, was leading to increased ternary complex affinity that would drive more efficient and possibly more rapid target ubiquitination and degradation. To measure the rate of protein degradation, we treated A375 cells with a single concentration of each PROTAC and measured the decrease in target abundance over time as shown in Figs. 4b and 5b. For each PROTAC, the concentration used was at or near the concentration where maximal activity was observed in dose response assays.

A positive correlation ($r = 0.67$) exists between $\alpha$ and initial SMARCA2 degradation rates (Fig. 4e). PROTACs **7** and **8**, possessing longer linkers, demonstrated negative cooperativity ($\alpha$) and lower initial SMARCA2 degradation rates. As discussed above, altering the SMARCA2 ligand (**9** or **10**) also led to negative cooperativity ($\alpha$) and lower initial SMARCA2 degradation rates. Conversely, PROTACs with modified linking vectors such as **1**, **2**, and **4** exhibited positive cooperativity ($\alpha$) and similar time course SMARCA2 degradation profiles (Fig. 4b) and initial SMARCA2 degradation rates. Considering the entire data set, initial SMARCA2 degradation rates demonstrated a negative correlation with ternary complex binding affinity ($r = -0.7$) and a positive correlation with cooperativity ($r = 0.67$) (Fig. 4e, f). Importantly, this work illustrates the need to track both ternary complex affinity and cooperativity. Though compound **6** (AU-15330) promotes high ternary complex affinity, it also exhibits relatively modest cooperativity ($\alpha = 2$). Cooperativity is calculated by taking the ratio of a PROTAC's binary to ternary complex binding affinities. Because compound **6** (AU-15330) has shown higher binary binding affinity, PROTAC to VHL ($K_{LP} = 11$ nM), relative to other PROTACs in the series (Table 3), the calculated cooperativity is also lower relative to PROTACs with similar ternary complex binding affinity (i.e., compound **1**). Despite these unique deviations from the trend, the SMARCA2 PROTAC series shows that positive cooperativity promotes higher ternary complex affinity, higher potency, and higher rates of target degradation.

BRD4 PROTACs exhibited a strong positive correlation ($r = 0.99$) between $BRD4^{BD2}$ cooperativity ($\alpha$) and initial BRD4 degradation rates (Fig. 5e). Increasing linker length diminishes the cooperativity ($\alpha$) and reduces the degradation rate. In contrast, there is no correlation between cooperativity and BRD4 degradation rate when $BRD4^{BD1}$ is

bound, reconfirming that $BRD4^{BD2}$ engagement is the key driver of BRD4 degradation[26,28,41]. Figure 5f shows the negative correlation ($r = -0.9$) between initial BRD4 degradation rates and ternary complex binding affinity.

### Relationship between ternary complex stability and target degradation kinetics

To better understand the role of ternary complex stability (i.e., half-life or dissociation rate constant) in influencing the target degradation rate (i.e., protein half-life), we analyzed SMARCA2 PROTACs, particularly compounds **1-5** with similar physiochemical properties. PROTACs **2**, **4** and **5** induced ternary complexes with fast dissociation rate constants compared to the slow dissociation rate constants exhibited by **1** (Fig. 4g). Consequently, **2**, **4**, and **5** exhibited short-lived ternary complexes compared to **1** (Fig. 4g, h). These four PROTACs have demonstrated similar initial rates of SMARCA2 degradation (Fig. 4b–h). A plot of all SMARCA2 degraders (Fig. 4h) demonstrated that there is a weak correlation ($r = 0.5$) between degradation rate and stability of the ternary complex. This observation suggests that slower dissociation rates or long-lived ternary complexes may not be the key driver affecting the degradation rate or half-life of SMARCA2 with this series of PROTACs.

It is worth noting that an analysis by Roy et al. previously reported the opposite conclusion involving degradation rate for a series of BRD4 degraders[28]. Taking this into account, we reanalyzed data surrounding our BRD4 degraders for ternary complex stability and degradation rate. The SPR sensorgrams in Fig. 5g indicate that ternary complex dissociation rate involving $BRD4^{BD2}$ increases with the increase in linker length compared to **17** (MZ1). Thus, except **17** (MZ1), other BRD4 degraders form progressively shorter-lived ternary complexes with $BRD4^{BD2}$. Figure 5h shows a strong correlation ($r = 0.95$) between the initial BRD4 degradation rate and the half-life of ternary complex, consistent with the previously published reports[28,42].

### Interfacial buried surface area of PROTAC-mediated ternary complex correlates with ternary complex binding affinity

To fully understand SAR surrounding our SMARCA2 degraders and develop new means to predict cooperativity and ternary complex stability in the future, we built a computational model for PROTAC-induced ternary complex formation and used molecular dynamic (MD) simulations to study the interfaces formed as a result of PROTAC binding. The simulated trajectories for the top model for each PROTAC-induced ternary complex were used to calculate the total buried surface area (BSA) at the interface[43–45]. Details of our model and simulations are discussed in the Methods. To verify the modeled ternary structure, we compared top models with published crystal structures for three different PROTAC systems including an in-house ternary crystal structure of compound **11** (PROTAC 2)[26,31,46]. As illustrated in Fig. 7a, overlay of in-house ternary complex crystal structure (PDB: 8G1P) of compound **11** (PROTAC 2) is similar to that of the published structure (PDB: 6HAX), with a RMSD <3 Å. Additionally, the BSA of modeled complex with compound **11** (PROTAC 2) is consistent with that calculated from both 8G1P and 6HAX[31] (Supplementary Information).

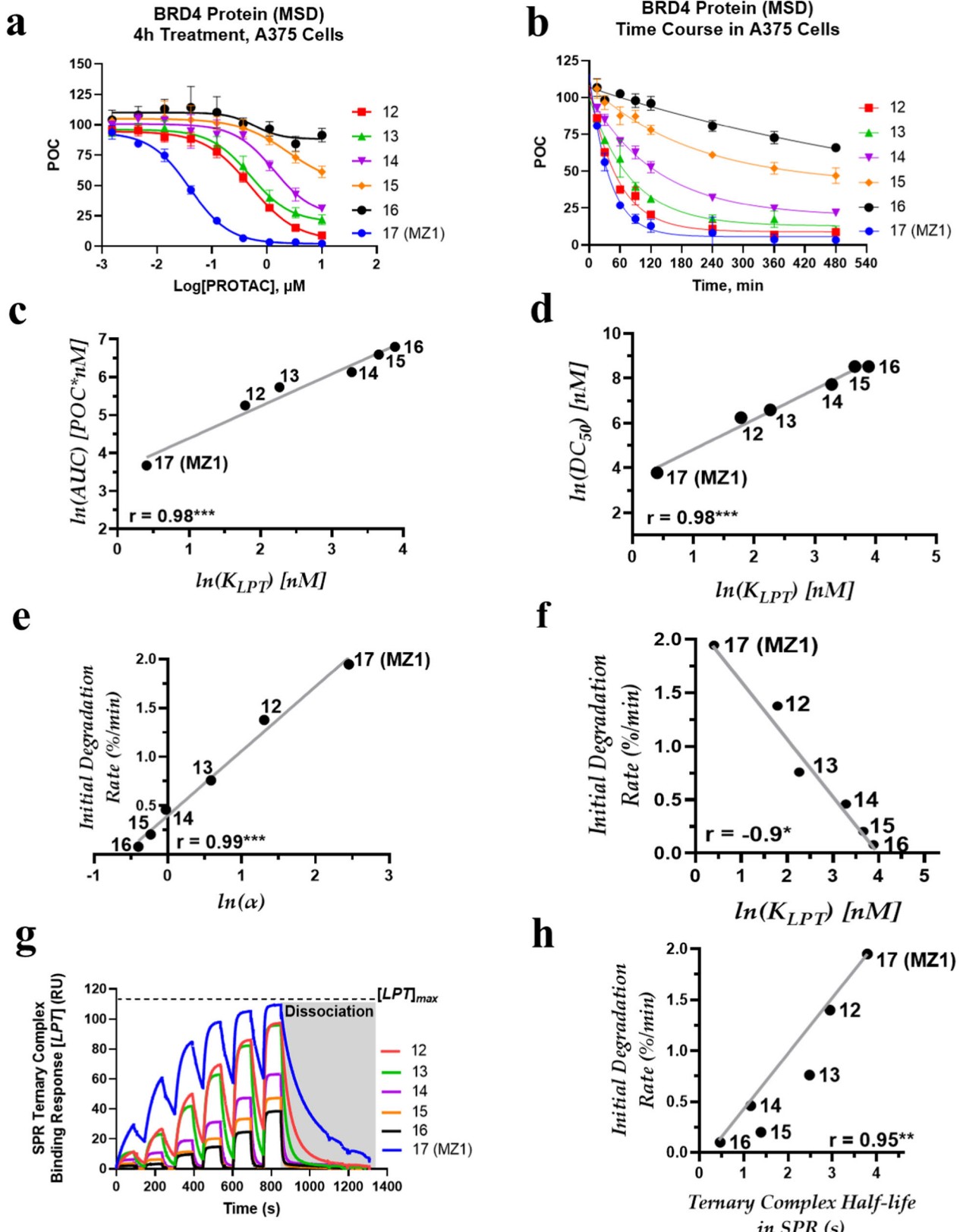

Attempts to solve the ternary complex crystal structure between VBC and SMARCA2$^{BD}$ with compound **1** was unsuccessful but we obtained a ternary complex crystal structure with its paralog SMARCA4 (Fig. 7b). We modeled the SMARCA2$^{BD}$ ternary structure with compound **1** using this crystal structure by substituting SMARCA4$^{BD}$ for SMARCA2$^{BD}$ and performed MD simulations to relax the system. The SMARCA2$^{BD}$-Compoud **1**-VBC complex appears to be more flexible than its SMARCA4 counterpart. The three top models of compound **1** induced ternary complex from the modeling workflow are all within the structural ensemble generated from simulating the crystal structure derived SMARCA2$^{BD}$-Compound **1**-VBC system (Supplementary Fig. 11).

**Fig. 5 | Relationship between ternary-complex attributes and degradation parameters of BRD4 degraders. a** Dose-response curve based on MSD assay for BRD4 PROTACs in A375 cells. **b** PROTAC-induced degradation time course in A375 cells. Error represents SEMs for 3 biologically independent experiments in 4 h time point MSD assay. Time course curves are a best fit (one-phase decay) of means from 3 biologically independent experiments, error represents SEMs. Correlation between PROTAC's **c** area under the dose–response curve (AUC) and **d** half-maximal degradation concentration (DC$_{50}$) against ternary complex binding affinity ($K_{LPT}$) for BRD4$^{BD2}$. Both AUC and DC$_{50}$, demonstrate strong correlations with $K_{LPT}$ with ***$P = 0.0004$ and ***$P = 0.0001$, respectively. Initial target degradation rate is plotted against **e** cooperativity factor ($\alpha$) and **f** binding affinity ($K_{LPT}$) for BRD4$^{BD2}$. A strong positive correlation (***$P = 0.0001$) exists between initial degradation rates and $\alpha$. Conversely, $K_{LPT}$ show a negative correlation (*$P = 0.01$) with initial BRD4 degradation rates. **g** SPR sensorgrams of PROTAC-mediated ternary complex formation between VBC and BRD4$^{BD2}$ with a gray box depicting dissociation of ternary complex, and **h** graph displaying a strong correlation (**$P = 0.003$) between initial BRD4 degradation rate and half-life (*log*) of BRD4 PROTACs in SPR. Pearson correlation coefficient (r) is used for correlation analysis and two-tailed test for significance.

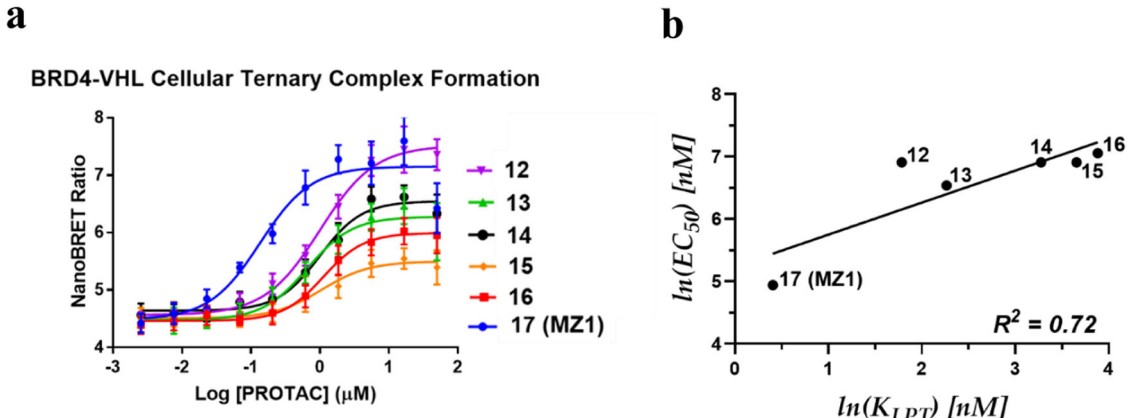

**Fig. 6 | Ternary complex formation in cells. a** Dose-response curve based on NanoBRET target engagement assay for BRD4 PROTACs. **b** Correlation between ternary complex binding affinities using live cells (EC$_{50}$) and purified proteins ($K_{LPT}$). Curves are a best fit of means +/- SD from 3 biologically independent experiments.

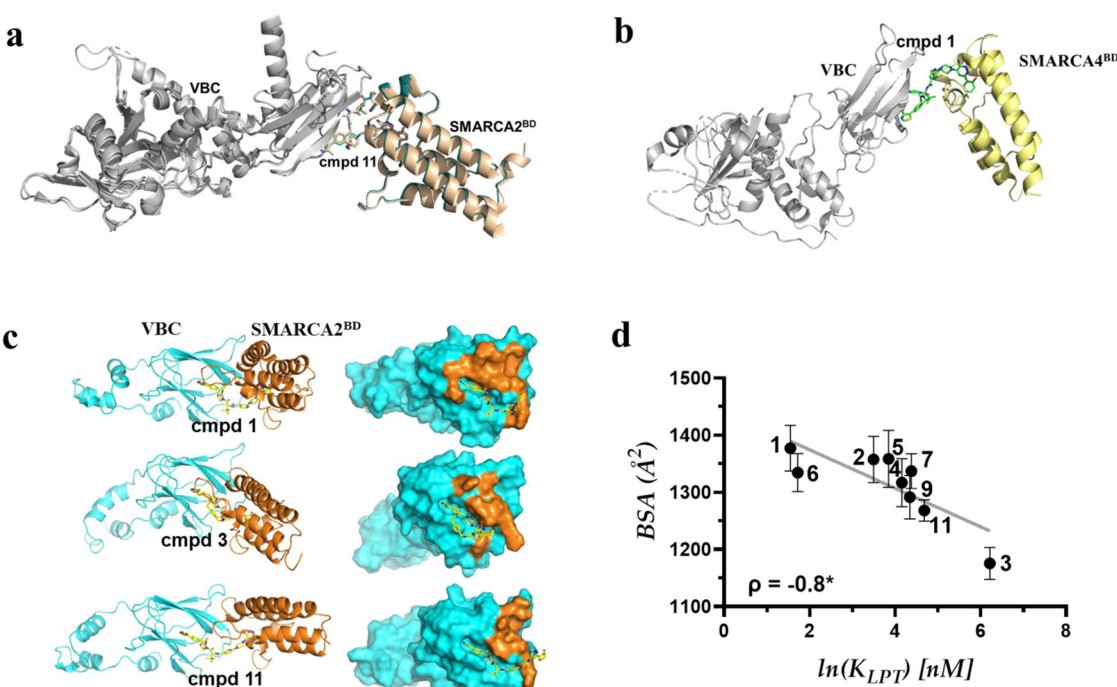

**Fig. 7 | Computational modeling of ternary complex. a** Overlay of ternary complex crystal structures (PDB IDs: 8G1P and 6HAX) of SMARCA2$^{BD}$/VBC with compound **11** in ribbon format with SMARCA2 in wheat (6HAX) and teal (in-house, 8G1P). **b** Ternary complex crystal structure (PDB ID: 8G1Q) involving SMARCA4$^{BD}$/VBC with compound **1**. Modeled PROTAC-induced ternary complex structures involving VBC and SMARCA2 bromodomain to calculate the buried surface area (BSA). **c** The left panel shows cartoon representation of the ternary complexes with VBC in cyan, SMARCA2 in orange, and PROTACs, **1**, **3** and **11** in yellow stick. On the right panel, surface presentation of VBC with PROTACs bound as in the ternary structure. The structure of SMARCA2 is removed for clarity, and the residues on VBC that are interacting with SMARCA2 are highlighted in orange. **d** Scatter plot for correlation between PROTAC-induced ternary complex binding affinity ($K_{LPT}$) and BSA. Spearman rank coefficient ($\rho$) is used for correlation analysis and two-tailed test for significance; error is SEMs for $N = 3$ top poses of the ternary complexes; * indicates $P = 0.014$.

Computational modeling revealed that the total BSA of compound **1** is much higher than its regioisomer, **3**, or a benchmark **11** (PROTAC 2)[31] (Fig. 7c/7d). None of the initial models generated for compound **8** pass the model selection criteria and therefore were not included in Fig. 7d. Calculated BSA was evaluated by correlating with the experimental binding affinity ($K_{LPT}$) data and analyzed using Spearman's ranking correlation coefficient ($\rho$)[47]. As shown in Fig. 7d, BSA demonstrated strong negative correlation ($\rho = -0.8$) with $K_{LPT}$, indicating that molecules with higher ternary complex binding affinities (lower $K_{LPT}$) have larger BSA. A correlation between the total BSA and $K_{LPT}$ is not surprising, given that these compounds have the same SMARCA2 and VHL ligands and the newly formed interactions upon ternary complex formation are largely non-specific. Under such conditions, the total interacting surface area roughly represents the stability of the ternary complex. Additionally, the positive correlation indicates that desolvation is likely the driving force behind the PROTAC induced ternary structure formation between SMARCA2 and VBC.

All analogs with positive cooperativities likely benefit from enhanced protein-protein interactions. Increasing the linker length of the PROTAC (i.e., **7**) could cause greater conformational freedom, and could impart flexibility in the ternary structure, resulting in reduced total BSA compared to compound **1** which lends more structural rigidity to the ternary complex. These results indicate that in the case of SMARCA2-VHL PROTACs, BSA can also be used as a surrogate to predict and rank the ability of the PROTACs to induce ternary complexes. It is important to note that here the comparison between BSA and $K_{LPT}$ is made within the same ligase-target system.

## Discussion

Strategies that leverage induced proximity to eliminate disease-causing proteins could bring many traditionally "undruggable" targets into the crosshairs of medicinal chemistry. Novel PROTACs that have appeared in the literature over the past half dozen years have surprised the research community with their ability to induce rapid degradation of their target proteins[15,48]. The early success of the PROTAC strategy has turned into a race to find ever more efficient degraders that possess the necessary properties required to become therapeutics. Though powerful, PROTACs are not a shortcut to small molecule drugs, and thus rational approaches to predict, measure, and guide SAR are needed.

How we evaluate the cellular activity of heterobifunctional small molecules and relate their activity back to biophysical properties is changing as the research community discovers new examples. Reports detailing the biophysical, structural, and cellular characterization of BRD4 degrader molecules[26,28,37] have established the central themes that will be repeated with new degraders. The work presented herein strives to build upon previous efforts by placing PROTAC-mediated ternary complex formation into a relatively simple mathematical framework that helps us to describe ternary complex formation with parameters derived from SPR experiments. In these studies, PROTAC potency and efficiency were described by calculating AUC from dose response curves and by calculating initial degradation rate from time-course degradation assays. We used PROTACs that target two different bromodomain-containing proteins that exploit the same E3 ligase, VHL, and postulated that the initial degradation rates might be correlated with the PROTAC's cooperativity based on the simplified ubiquitination kinetics relationship.

Our work demonstrates that ternary complex affinity ($K_{LPT}$) and cooperativity ($\alpha$) drive the cellular activity of degraders and these two parameters are linked but not equivalent. $K_{LPT}$ accounts for the sum of the interactions between all three components (Fig. 1a, ternary complex equilibria) during ternary complex formation, while cooperativity describes the effect of a binding partner (i.e., E3 ligase) on the PROTAC's interaction with the other binding partner (i.e., target protein),

either in a synergistic ($\alpha > 1$) or antagonistic ($\alpha < 1$) manner. The correlation between cooperativity and initial degradation rate for SMARCA2 degraders highlighted a difficult to predict but nonetheless critical attribute of our molecules. For a handful of structurally similar SMARCA2 degraders, tuning the linking vector between target and ligase ligands to maximize cooperativity may have also positioned substrate relative to the ligase with an orientation that is optimal for efficient ubiquitination and degradation. SMARCA2 PROTAC **1** demonstrated the highest cooperativity ($\alpha = 12.8$) compared to its closest analogs **2, 4** and **5**. The maximal ternary complex formation increases with cooperativity and reaches saturation, after which it becomes independent of the cooperativity factor (Fig. 2c). Compound **1** and its analogs have similar binary binding affinity parameters ($\beta \approx 0.2$), and $90\% \, [LPT]_{max}$ is reached at $\alpha = 2$. We reasoned that once the critical concentration of PROTAC-mediated ternary complex has been achieved, further increases in cooperativity may not impact degradation efficiency. As a result, the initial rate of SMARCA2 degradation is similar for all the close analogs of **1**.

Conversely, BRD4 degraders demonstrated that initial degradation rate is dependent on both cooperativity and ternary complex stability. The long half-life of MZ1-induced ternary complexes, ~130 s, promotes efficient target degradation in cells[26,28]. Throughout our series of BRD4 degraders, the rate of BRD4 degradation increases with the increase in half-life of ternary complex and cooperativity. We reasoned that higher cooperativity is required for MZ1 (**17**) to reach $90\% \, [LPT]_{max}$ due to the higher binding affinity parameter ($\beta \approx 0.4$) (Fig. 2c)[26] and longer ternary complex half-life promotes efficient ubiquitin transfer and target degradation[49].

To determine if ternary complex off-rate was responsible for the higher initial rate of degradation for our SMARCA2 degraders, we measured half-life for the series. Though the half-life of the ternary complex varied significantly for the closely related PROTACs there was not a strong correlation between complex half-life and initial degradation rate. This difference was somewhat surprising especially since the ternary complex induced by our most active PROTAC exhibited a half-life of just 40 s, 3-fold less than that demonstrated by MZ1. We can only speculate that the requirement for complex stability will vary between different substrate proteins and possibly between series of PROTACs for the same substrate and that for our most active SMARCA2 degraders, we have surpassed a threshold of induced ternary complex stability and any increases in stability no longer drive more efficient ubiquitination and degradation.

Measuring and following cooperativity impacted our design of SMARCA2 degraders in another profound way. Improving cooperativity allowed us to identify heterobifunctional molecules that have reduced molecular weight and characteristics that facilitate their use in pre-clinical in vivo models. The identification of cooperative molecules allows for the design of degraders with lower affinity ligands to both target protein and ligase, along with some room to optimize the pharmacokinetic properties. Additionally, this study demonstrated that one parameter cannot sufficiently describe the degradation activity of a PROTAC. For example, compounds **2, 4** and **5** have shown similar SMARCA2 degradation rates compared to **1**, albeit their AUCs (potency) are inferior. For these VHL-dependent degraders, high ternary complex binding affinities and positive cooperativity correlated with degradation potency (AUC) and initial rate of target degradation. Ternary complex formation and induced protein degradation can be inhibited by excess PROTAC leading to the frequently observed "hook effect". The observed squelching of activity at high concentrations of degraders is caused by the unproductive binary interaction of the PROTAC with substrate or ubiquitin ligase rather than a productive ternary complex of both target and ligase. It has been previously observed that PROTACs that promote ternary complexes from cooperative protein-protein interactions suffer less from the "hook effect"[50,51]. In this study, we similarly observe (Fig. 4a)

reduced "hook effects" for those molecules (i.e., **1, 2**, and **4**) that promote the greatest cooperativity.

At the start of a PROTAC discovery campaign when structural information is limited, identifying a lead series and optimizing toward cellular activity is not straightforward. Crystallization of PROTAC-mediated ternary complexes is especially challenging at this stage. We and others[45] have found that getting diffraction quality crystals of ternary complexes can be hit-or-miss. For example, despite our numerous attempts we were unable to grow crystals for many of our own SMARCA2 degraders but solved the structure of the previously reported[31] SMARCA2-Compound **11**-VBC complex to 2.7 Å (PDB: 8G1P). Additionally, we solved the structure of SMARCA4 bromodomain complexed with compound **1** and VHL at a resolution of 3.7 Å (PDB: 8G1Q). The SMARCA4 bromodomain is highly homologous to the SMARCA2 bromodomain[52], thus this lower resolution structure provided a valuable starting point for MD simulations and showed agreement between the top models of SMARCA2$^{BD}$-Compound **1**-VHL from the ternary structure modeling workflow and the simulated structural ensemble (Fig. 7b and Supplementary Fig. 11).

This work was motivated by the desire to develop a predictive framework based on measurable SPR parameters rather than crystallographic data of induced ternary complexes. Herein, total BSA calculated from the top model of the ternary complex correlates well with binding affinity and can be used to evaluate the predicted SAR of PROTACs with varying architectures (i.e., linking vector and length) prior to synthesis. Confidence in computational modeling grows as a greater number of PROTAC molecules are tested in both SPR and cellular degradation assays, which in turn will guide novel design strategies. Extending the SPR and cell-based methods presented in this manuscript to other ligase-target pairs[53,54] will broaden our growing appreciation of the unique SAR that arises during the engineering of PROTACs. We hope that advances in structure-based computational modeling[55], free-energy perturbation (FEP) methods[56] and cryo-electron microscopy to probe the conformational states of ternary complexes[57], will make PROTAC design more efficient.

## Methods
### Cellular degradation assay
A375 cells were treated with SMARCA2 or BRD4 degraders for 2 h (4 h for BRD4) and generated lysates. SMARCA2 or BRD4 protein was measured with a variation of a sandwich ELISA that was assembled from commercially available antibodies as well as reagents and instrumentation procured from Meso Scale Discovery (MSD). Degradation parameter, DC$_{50}$, is not suitable for PROTACs that induce partial/incomplete DR curves at a fixed time point; maximum drop in target protein level (D$_{max}$) >>0. Conversely, the area under the curve (AUC) can be calculated as a degradation activity parameter for any DR curve, and it encompasses both the degradation potency (i.e., DC$_{50}$) and efficacy (i.e., D$_{max}$) of PROTACs.

**Cell culture.** A375 cells (ATCC® CRL-1619™) were cultured in RPMI 1640 Medium (ThermoFisher Scientific 11875093) containing 10% fetal bovine serum (ThermoFisher Scientific 16000044) and 1x penicillin-streptomycin-glutamine (ThermoFisher Scientific 10378016). 16 h prior to compound treatment, cells were seeded in 96-well cell culture plates (Corning 3904) at a density of 2.5 ×10$^5$ cells/well (90 µL/well) and incubated at 37 °C, 5% CO$_2$. A 1:3 compound dose-response titration was diluted in growth media (1:20), added to appropriate wells of a cell culture plate (1:10) and then assay plates were incubated at 37 °C, 5% CO$_2$. After 2 h of compound treatment, cells were lysed in MSD lysis buffer (MSD R60TX-2) containing protease (Roche 04693116001) and phosphatase (Roche 04906837001) inhibitors.

**Protein detection.** MSD standard binding plates (MSD L15XA-3) were coated with 40 µL of 2 µg/mL SMARCA2 (Active Motif 39805) capture antibody overnight at 4 °C. Plates were then incubated on a plate shaker with 150 µL per well 3% BSA (MSD R93BA-4) for 1 h, 25 µL per well of cell lysates for 1 hr, 25 µL per well of SMARCA2 (0.25 µg/mL Cell Signaling 11966 S) detection antibody for 1 h then 25 µL per well of 0.5 µg/mL Sulfo-tagged rabbit (SMARCA2, MSD R32AB-5) for 1 h. Plates were washed with 300 µL per well MSD wash buffer (MSD R61TX-1) between each step. Following the last incubation, plates were washed with MSD wash buffer then 150 µL of MSD read buffer (MSD R92TC-2) was added to each well. Plates were immediately read on a MSD plate reader (MSD Sector Imager 6000).

**Data analysis.** MSD electrochemiluminescence signals were subtracted by the average background signal from wells with lysis buffer alone. Normalized MSD values (POC) for individual compound treated wells were acquired by dividing background-subtracted assay signal by signal from vehicle control wells (cells + 0.1% DMSO) and multiplying by 100. This POC data was graphed using a nonlinear regression curve fit xy analysis and a log(inhibitor) vs response −variable slope (four parameter) model for dose response curves (Graphpad Prism 9, San Diego, CA, USA). To calculate area under the curve, all points on the curve were connected and vertical lines were drawn from each point down to $y = 0$, generating a series of trapezoids. The area of each trapezoid was determined using the equation $A = (1/2)(h)(b1 + b2)$, where $h = 1$ (distance between points), b1 = POC of the first point and b2 = POC of the second point. The sum of all trapezoids provides an approximate area under the curve (AUC). For half-life (t$_{1/2}$) determination in time course studies, a one-phase decay model was used for curve fitting and the time at which the curve crossed 50 POC was determined through interpolation (Graphpad Prism 9, San Diego, CA, USA). Additionally initial rate of degradation was determined by multiplying 100 with rate constant ($\tau = 0.693/t_{1/2}$).

### Protein purification
Purified proteins VBC, SMACRC2$^{BD}$ (bromodomain), BRD4$^{BD1}$ or BRD4$^{BD2}$ were expressed and purified according to the previously published protocols[28,31]. Briefly, purification of N-terminal His$_6$ tagged proteins (1) SMARCA2$^{BD}$ 1373–1511 Δ1400–1417; (2) VHL (54-213), ElonginB (17-112) ElonginC (1-104); and (3) BRD4 (44-168) cells were lysed by 2 passes on microfluidizer and ultracentrifuged at 235,000 g for lysate clarification. Clarified lysate was mixed with Talon resin and nutated for an hour at 4 °C, washed and eluted with 250 mM imidazole. The His$_6$ tag was removed using TEV protease. Cleaved proteins were then purified using Superdex S-75 column (GE Healthcare). Purified protein samples were stored in 10 mM HEPES 7.5, 150 mM NaCl, 0.5 mM TCEP at -80 °C to be used for crystallization/SPR studies.

### SPR binding studies
SPR experiments were performed on a Biacore 8 K or T200 instrument (Cytiva). Immobilization of His-tag VHL/EloB/EloC (VBC) was carried out at 25 °C using Series S NTA chip, where VBC was first captured via His/NTA affinity followed by amine-coupling using NHS/EDC, and finally deactivation of the unfunctionalized carboxy groups using 1 M ethanolamine. The running buffer during immobilization was PBS with 0.005% P20. Either high-density (3000-4000 RU) or low-density (300-400 RU) VBC-functionalized surfaces were created, followed by equilibration in the running buffer for 3 h.

PROTACs (10 mM stocks in 100% DMSO) were diluted to 400 nM in a running buffer (20 mM Tris pH 7.5, 200 mM NaCl, 0.02% P20). This stock solution was then serially diluted in the running buffer containing final 2% DMSO. Solutions were injected in multi-cycle kinetic format without regeneration (contact time 60 s, flow rate 80 µL/min, dissociation time 120 s) using a stabilization period of 30 s and syringe wash (50% DMSO) between injections onto a high-density VBC-functionalized surface. For SMARCA2 analysis, biotinylated proteins were introduced on a Series S streptavidin (SA) sensor chip at high-density

(3000-4000 RU). Conversely, for BRD4 binary interaction, His-tagged protein was introduced on a Series S NTA chip, where BRD4 was captured/coupled via His/NTA affinity followed by amine-coupling, and finally deactivation of the unfunctionalized carboxy groups using 1 M ethanolamine.

PROTACs (10 mM in 100% DMSO) were initially prepared at 200 nM in a running buffer (20 mM Tris pH 8.0, 250 mM NaCl, 1 mM TCEP, 0.01% P20, 0.2 mg/ml BSA) with a concentration of 2% DMSO. This solution was mixed at 1:1 ratio with a solution of 5 μM of the SMARCA2 or BRD4 bromodomain protein in the running buffer. This complex was then serially diluted in the running buffer containing 2.5 μM SMARCA2 or BRD4 and 2% DMSO (5-point five-fold serial dilution). For ternary experiments, solutions were injected sequentially in single-cycle kinetic format without regeneration (contact time 60 s, flow rate 80 μL/min, dissociation time 300 s) using a stabilization period of 60 s and syringe wash (50% DMSO) between injections onto a low-density VBC-functionalized surface.

Raw sensorgrams were processed by performing double reference subtraction, solvent correction, and analysis was done using Biacore Insight Evaluation Software. Kinetic analysis was performed by fitting data to a 1:1 Langmuir interaction model and a steady-state affinity model was used to evaluate equilibrium binding affinity.

### Buried surface area calculation

**PROTAC induced ternary structure modeling**. An internal PROTAC induced ternary structure modeling workflow was developed. This workflow is similar to the published methods in that it utilizes linker conformational search and protein-protein docking to produce ternary complex models[58–60]. The difference of the method lies in that it takes into account linker strain energy in selecting top models as well as relies on all-atom molecular dynamics (MD) simulations to relax the final models. The procedure of the model generation process is given below.

**Step 1: PROTAC fragment conformational search.** The ligand bound VHL structure (PDB ID: 4W9H)[61] is used as a starting point for generating PROTAC fragment ensemble. The coordinates of the VHL ligand are kept while the linker is added to the ligand structure with an arbitrary conformation. The linker is defined as a portion that belongs to neither the SMARCA2 nor the VHL ligand. SMARCA2 ligand is partially appended to the linker to form the fragment used in the conformational search for structural assembly in the subsequent steps. Schrodinger MacroModel conformational search tool[62] is used to perform restrained PROTAC fragment conformational search. The default implicit solvent model with water as solvent was chosen to solvate the systems. Mixed torsional/low-mode sampling was used to search the conformational space with 1000 steps per rotatable bond. Each step was minimized with the OPLS3e force field[63] using the Polak-Ribier Conjugate Gradient (PRCG) method with maximum iterations of 5000 and energy convergence threshold of 0.05. The heavy atoms on the VHL ligand are frozen in place to avoid unnecessary sampling of the VHL ligand conformation. All resulting structures within 20 kcal/mol of the lowest energy conformation were saved.

**Step 2: Ensemble protein-protein docking.** To capture conformational changes upon complex formation, ensemble protein-protein docking was used in producing complex models with reasonable protein-protein interfaces. The docking input conformations of the monomer proteins were generated using MD simulations starting from the ligand bound SMARCA2 (PDB ID: 6HAZ)[31] and VHL (PDB ID: 4W9H)[61]. The all-atom systems were built using the system builder application in Schrodinger with 10 Å padding in all three directions. The system was neutralized with counter ions and TIP3P water was added along with 0.15 M KCl to solvate the system. Desmond (Association for Computing Machinery, Tampa, Florida; 2006) was used to

generate 240 ns simulation trajectories for each system. Customized OPLS3e force field was used in all simulations. The last half of the simulation trajectories were sampled in 12 ns interval to produce 10 structures for SMARCA2 and VHL each for the ensemble docking step. The PIPER program[64,65] in Schrodinger was employed to perform protein-protein docking calculations. Repulsive restraints were applied on residues that are >15 Å away from the ligand binding sites. 1000 decoys were generated for each input pair resulting in a total of 100,000 decoys.

**Step 3: Structural assembly.** The ten each, ligand bound SMARCA2 and VHL structures used as protein-protein docking inputs were included in assembling preliminary complex models. Each structure is superposed to a fragment conformation using three shared anchoring atoms. Once both the SMARCA2 and VHL structures were superposed to a PROTAC fragment conformation, number of clashes between the two proteins and between the proteins and the PROTAC were computed using a distance cutoff of 2.2 Å. The generated models were only kept if the number of clashes is less than or equal to 5 for protein-protein interactions and 2 for protein-PROTAC interactions. These numbers are empirical and can be tuned to include more models for the next model pose pruning step.

**Step 4: Model pruning.** Since protein-protein interactions are important in ternary structure formation, the information gained from protein-protein docking needs to be incorporated into the modeling selection process. This is achieved through comparing the models generated by Step 3 and the docking decoys generated in Step 2. To simplify the comparison, we aligned all the complex models using the VHL structure and used the heavy atoms in the SMARCA2 ligands (SMILE: c1cccc(O)c1-c2cccnn2) to indicate the position of the SMARCA2 protein. Noticing the elongated shape of the SMARCA2 ligands, we decided to further reduce the dimension of the data by representing the SMARCA2 ligand using its center of geometry (COG) and the three principal components. The difference between the models from Step 2 and Step 3 was computed as the distances between the COGs and the SMARCA2 ligands as well as the angles between the first and the second principal axes respectively. Models with COG distances <6 Å and the deviation in the first and the second principal axes <20° were considered similar and kept for further refinement.

**Step 5: Further filtering using linker conformational strain.** Similar to small molecule binding to proteins, the conformational strain of a PROTAC upon binding to form ternary complexes should be relatively small. This is the rationale behind using linker strain energy as a filter to further narrow down the models generated. First, the force field parameters were generated for the entire PROTAC molecule using the Force Field Builder tool in Schrodinger. Afterwards, the linker conformations were extracted from the models after Step 4 and subjected to energy calculations with the customized OPLS3e force field. A restraint-free conformational search on the linker was performed using the same parameters as in Step 1 except that no positional constraints were used. A ground state conformation was identified as the one with the lowest OPLS3e energy in the conformational search and the energy of this conformation was set as the reference energy. The strain energy of the linker was defined as the difference between the calculated and the reference energies. A cutoff of 4 kcal/mol was used to retain models. It is worth noting that quantum mechanical (QM) calculations can be used to determine the linker strain energies more accurately. However, due to its resource-intensive nature, force field energies were used here.

**Step 6: MD simulations to relax the models.** As the models were generated using uncorrelated components, it is important to perform relaxation and generate data for statistical analysis. MD simulations

were used for both purposes. The simulation protocol was the same as the in Step 2. For each candidate model, a 120 ns simulation trajectory was generated in saved in the form of 1000 frames. The last 20% of the trajectories were used to compute properties such as energies and buried surface areas.

**Model selection and BSA calculation.** After MD simulations, we used the trajectories to further interrogate the validity of the generated models. A few properties were computed including linker strain energies, PROTAC target warhead root mean squared deviation (RMSD) and total buried surface area (BSA). The buried surface area was computed as the sum of protein-protein BSA and the protein-PROTAC BSA. To calculate the BSA, the solvent accessible surface area (SASA) of each individual component was first computed followed by the SASA of the complex. The difference between the sum of individual SASAs and the complex SASA was designated as the BSA. The final BSA for each PROTAC induced ternary complex was computed as the BSA average using trajectories of model systems that have passed the selection criteria, which was defined by linker strain energy within thermal fluctuation (i.e., 0.6 kcal/mol) and the RMSD of the target warhead heavy atoms <3 Å. Using the same protocol, we were able to successfully model known ternary structures induced by VHL-based PROTACs (Fig. S9). The near-native structures were found within top three of all models generated for all three systems (i.e., VHL/PROTAC2/SMARCA2). With this finding, we decided to apply the modeling workflow to evaluate the ternary structure formation capabilities of the PROTACs presented in this manuscript.

### Ternary complex crystallization

Crystallization of PROTAC-mediated ternary complex was performed using sitting drop vapor diffusion method on 96 well trays. VBC:SMARCA2$^{BD}$ were mixed as a 1:1 stoichiometric ratio in 10 mM HEPES (7.5), 150 mM NaCl, 0.5 mM TCEP and concentrated to ~10 mg/ml. Compound **11** (PROTAC 2) and compound **1** was then added to mixture in 1:1 stoichiometric ratio and incubated for 20 min in ice. Drops of the ternary complex were mixed 1:1 and 1:2 in crystallization buffer using a Mosquito® robot (SPT Labtech). With compound **11**, crystals appeared within 7 days in reservoir solution containing 20% PEG 3350, 0.2 M sodium chloride.

Extensive screening efforts with compound **1** with SMARCA2$^{BD}$ and VBC failed to generate any hits. HT screening efforts with compound **1** with SMARCA4$^{BD}$ and VBC resulted in hits in reservoir solution containing 0.1 M BIS-TRIS pH 6.5, 25% of polyethylene glycol 300. These crystals were then optimized for crystal growth using streak seeding method. Crystals were flash frozen in reservoir solution supplemented with glycerol as cryo-protectant. All data sets were collected on a Pilatus3 6 M silicon pixel detector at the Advanced Light Source Beamline 5.0.2 at wavelength 1.00000 Å and temperature 100 K. The data were integrated and scaled using HKL2000. The structures were solved by molecular replacement using Phaser from the CCP4 program suite with apo-SMARCA2 and apo-VBC as a search model. The structures were refined using Phenix. The structure of VBC:Compound **11**: SMARCA2$^{BD}$ refined to 2.7 Å resolution with R-factor of 21% and R$_{free}$ of 26%. The structure of VBC: Compound **1**: SMARCA4$^{BD}$ refined to 3.7 Å resolution with R-factor of 23.4% and R$_{free}$ of 32.5%. Coordinates for the structure of SMARCA4-Compound **1**-VBC and SMARCA2-Compound **11**-VBC have been deposited to the PDB with the accession codes 8G1Q and 8G1P, respectively.

### NanoBRET cellular ternary complex formation assay

HiBiT-BRD4 KI HEK293(LgBiT) cells (HiBiT-fused BRD4 to its N-terminus by CRISPR-Cas9 in HEK293 cells stably expressing LgBiT; Promega CS302312) were grown in DMEM (Corning 10-013-CV) supplemented with 10% FBS (Corning 35-010-CV). HiBiT-BRD4 KI HEK293(LgBiT) cells were transiently transfected with the HaloTag-VHL fusion vector using

FuGENE HD Transfection Reagent (Promega E2312). The HaloTag-VHL fusion vector contains the coding region for an N-terminal HaloTag fusion to VHL (Promega N2731). 18 h to 24 h post-transfection, the transfected cells were resuspended in Opti-MEM (Life Technologies 11058-021) containing 4% fetal bovine serum with 0.1 μM HaloTag NanoBRET 618 Ligand (Promega G9801), seeded into 384-well TC-treated ProxiPlates (PerkinElmer 6008239) at a density of 5 ×10³ cells/well (10 μL/well), and incubated at 37 °C, 5% CO$_2$ for assay the following day. 30 min prior to compound treatment, cells were preincubated with a proteasome inhibitor, MG132 (Selleck Chemicals S2619) at 10 μM at 37 °C, 5% CO$_2$. Then, a 1:3 compound dose-response titration was directly added to appropriate wells of a cell culture plate (1:200) using Echo 555 Liquid Handler (Labcyte Inc), and then assay plates were incubated at 37 °C, 5% CO$_2$. After 2 h of compound treatment, plates were equilibrated to RT for 15 min, followed by the addition of 5 μL 3x NanoBRET Nano-Glo Substrate (Promega N1572) in Opti-MEM. Plates were incubated at RT for 5 min and read on EnVision (PerkinElmer) equipped with the NanoBRET optics (PerkinElmer 2100-8530). BRET signals were measured within 1 h after adding substrate with 1.0-s integration time. BRET ratio (milliBRET unit) was defined as the ratio of acceptor signal over donor signal multiplied by 1,000, and data was analyzed using a 4-parameter logistic model to calculate EC$_{50}$ values.

### Compounds syntheses and characterization
Included in Supplementary Information

### Reporting summary
Further information on research design is available in the Nature Portfolio Reporting Summary linked to this article.

## Data availability
All data supporting the findings of this study are available in the main text and Supplementary Information file. Coordinates for the structure of SMARCA4-Compound **1**-VBC have been deposited to the PDB with the accession code 8G1Q. Coordinates for the structure of SMARCA2-Compound **11**-VBC have been deposited to the PDB with the accession code 8G1P. Additional information is available from the corresponding authors upon a request.

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

## Acknowledgements

We thank Rati Verma, Chris Fotsch, Ryan Potts for their suggestions in the preparation of this manuscript. Ray Deshaies, Philip Tagari, Margaret Chu-Moyer, Peter Hodder, and Peter Grandsard for their support of this program.

## Author contributions

A.V. conceived this study with contributions from D.M., R.P.W. and H.R. R.P.W., A.A., N.C., X.L., and A.B. designed and synthesized PROTACs. H.R. performed computational modeling and S.G.-R carried out VBC/ SMARCA2/BRD4 expression, purification, and crystallization. A.V. designed and performed SPR experiments with contributions from Q.C. K.C. performed NanoBRET target engagement and cellular permeability assays. K.D. and K.S. performed cellular degradation assays. A.V. analyzed and interpreted data with contributions from D.M. and K.D. A.V. and D.M. wrote the manuscript with contributions from R.P.W. and H.R.

## Competing interests

All the authors are employees and shareholders of Amgen, Inc.
