## [Peer Review File · Nature Communications]

Reviewers' Comments:

Reviewer #1:

Remarks to the Author:

The manuscript by A. Vaish and co-worker on "Affinity and Cooperativity Modulate Ternary Complex Formation to Drive Targeted Protein Degradation" presented experimental results and analyses that supported the correlation between PROTAC ternary complex affinity/cooperativity and the protein degradation efficiency through the studies of two different VHL-based PROTAC series. The manuscript was logical developed and well written that can be published as it is. The results and conclusions were consistent with known published work primarily by A. Ciulli's lab (references 26 and 27). However, they did not provide new insights into the relationship between the ternary complex formation and degradation. Even the "total buried surface" is not a new concept to characterize ternary complexes (for example, R. P. Law et al. *Angew. Chem. Int. Ed.* 60, 23327-23334 (2021)). The authors' work lacks the originality and impact to the field to be qualified for publication in *Nature Communications*. To further advance the knowledge for PROTAC design, the studies of non-cooperative ternary complex systems that lead to degradation would be more interesting (for example, M. F. Calabrese et al. *Nat. Chem. Biol.* 17, 152-160 (2021)).

Reviewer #2:

Remarks to the Author:

This is an excellent paper by a strong team at Amgen, studying PROTAC ternary complexes formation, affinity, cooperativity and kinetics stability and attempting to correlate parameters to the Kinetics of protein degradation. It does so by utilizing the well-characterized systems of Brd4-VHL (MZ1) and SMARCA2-VHL (ACBI1) previously reported and structurally and biophysically described by our Laboratory.

The authors develop a clever mathematical model for considering the equilibria in the SPR binding assay previously described (Roy et al. 2019). The parameter beta, reporting on the relative binary binding affinities is useful and relevant. They then characterise a library of 11 SMARCA2 PROTACs (including two benchmark literature compounds, compound 6 (AU-15330) and compound 11 (ACBI1)) and 6 BET PROTACs (including compound 17 ie. MZ1 and five triazole containing compounds, following their previous work on click-chemistry library generation (Wurz et al. 2018)).

The main finding of the paper is that the authors find that, broadly at least for the SMARCA2 degraders, degradation potency correlated with ternary complex affinity, while the initial rate of degradation correlated more with cooperativity. For the BRd4 degraders, they find the same as above, however (consistent with previous finding in Roy et al.) they in this case also saw that the initial rate of degradation correlated with the ternary complex half life ($t_{1/2}$) measured by SPR (although they do not plot this).

Finally, they model the SMARCA2 PROTAC ternary complex via MD simulations and find that the buried surface area (BSA) correlates with ternary K_d .

I am in two minds about this paper. On the one hand, it builds on extensive prior literature and in the most validates and corroborates previous findings that are now well established (I may have a biased view here, since this is what my laboratory has discovered and what have been the key messages of my lectures for several years now). I also feel that some of the correlations claimed are rather stretched given the poor correlation (R^2 0.6-0.7) and the low correlative power given the relatively few data points.

On the other hand, the work is for the most quite solid, and it contains areas of innovation and novelty - mainly, the mathematical treatment of the SPR equilibria, and the correlative analysis of the nicely quantitative measurements carried out. It also contains nicely designed chemical series of PROTAC compounds, including appropriate benchmark literature compounds, and it addresses important aspect of understanding and predicting SAR in PROTAC mode of action, that is a critical aspect of great interest in current degrader drug design campaigns. The paper is also well written, well illustrated and well referenced.

Overall, I would be supportive of publication in Nat Commun. should a revised version of the paper satisfactorily address the below major comments and concerns, that are aimed at constructively critique the work, and for the authors consideration with a goal to improve its scholarly contribution and placing in the context of the field.

1. Lines 253-255:

"17 (MZ1) exhibited a faster ternary complex dissociation with BRD4BD1 compared to BRD4BD2 (Figure 5C), and ternary complex binding affinity with BRD4BD1 is 10-fold weaker compared to BRD4BD2 (Table 1)."

These findings are entirely in agreement with those published by us previously (Gadd et al. 2017; Roy et al. 2019; Klein J Med Chem 2021 <https://doi.org/10.1021/acs.jmedchem.1c01496>) which warrant due references and citations here.

2. Same for lines 290-291:

"reconfirming that BRD4BD2 engagement is the key driver of BRD4 degradation." This needs citations to the above references, as per comment above

3. Wrt to correlating BSA to ternary complex stability - this has been discussed and scholarly treated before [Kozicka & Thoma, Cell Chem Biol 2021, <https://doi.org/10.1016/j.chembiol.2021.04.009>; and Cowan and Ciulli, Annu Rev Biochem, <https://doi.org/10.1146/annurev-biochem-032620-104421>] and hence the authors might want to duly reference these articles.

4. Fig. 2A and 2D seem to have overlooked / ignored the hook effect at high PROTAC concentration? The hook effect with the PROTACs studied in this work is clearly experimentally observed in the data shown in Fig. 4A, for example. This should be clarified and discussed

5. Figure 3A (should this not a Table instead?) seem to report Dmax as % of remaining protein after degrader treatment. This is conventionally reported as % of degraded (depleted) protein instead please revise

6. Please show the chemical structure of compound 11 in a main text figure / table, and specify that it is aka ACBI1. Equally, please specify that compound 6 is aka AU-15330

7. All BET PROTACs used in this study contain a triazole from the click chemistry, while MZ1 does not. So in addition to having longer linkers, they most likely also will have poorer cell permeability (a known limitation of triazole groups). This is seen as striking drop in degradation potency between MZ1 and all the other compounds (Fig. 5A), as well as striking drop in potency in cellular TE assay (Fig. 6B). This should be discussed. As the authors hint to, this is also reflective of the poor correlation ($R^2=0.72$) in Fig. 6C, as the cellular TE potencies of compounds 12-16 are all skewed to weak binding affinity likely due to poor cell permeability. The authors might want to perform the experiment again in permeabilized format (see Riching et al. Current Res Chem Biol 2021 <https://doi.org/10.1016/j.crchbi.2021.100009>; Imaide et al. Nat Chem Biol 2021 <https://doi.org/10.1038/s41589-021-00878-4>). Actually, in deeper inspection, it seems based on the last page of the Supporting Information that they have done this already and so have the data?

8. Table 1: please explain how AUC was calculated from the degradation assay data

9. Fig.s 4B and 5B. For the time course experiments, time-course profiles and hence $t_{1/2}$ will be dependent on PROTAC concentration. The authors should explain the criteria followed to select the PROTAC concentration used in these experiments

10. Figure 4E/F and 5E/F: what would the correlation plot look like if the authors plotted $\text{Log}(DC_{50})$ vs $\text{log}(K \text{ ternary})$? What would the correlation plot look like if they plotted degradation potency vs $\text{log}(\alpha)$, and conversely degradation rate vs $\text{log}(K \text{ ternary})$? For sake of fair comparison, and to allowed the reader to best judge, the authors should show those missing correlation plots too.

11. Fig.s 4F and 5F: what would the correlation plot look like if they plotted degradation rate vs

ternary complex SPR $t_{1/2}$? Based on their results discussion, a good correlation is seen for this relationship too, so the actual plot data should be shown to support the conclusion.

12. The differential correlative trends observed between SMARCA2 and Brd4 degraders are interesting, and the authors attempt to speculate on why that might be. One main difference between these two class of PROTACs is that the BET ligand has much higher binding affinity for the Brd4 bromodomain (K_d 10-100nM) than the SMARCA2 ligand which has K_d of 200-2000 nM) for the SMARCA2 bromodomain. The authors only measure binding affinities to the VHL ligase (K_{LP}) but do not measure the K_d for the bromodomains (K_{TP}) - and they might wish to include such measurements. I am wondering if these major difference could play a role in the observed differences in correlative trends?

I agree to waive anonymity in a spirit to enhance the transparency of the peer review process, Alessio Ciulli

Reviewer #3:

Remarks to the Author:

Unlike the conventional inhibitors, the pharmacology of PROTACs involve additional steps to take effect, i.e. the formation of the ternary structure, the recruitment of ubiquitin and hence the degradation, in addition to the binding of the inhibitor and target. Therefore, this makes it more difficult to 1) define the binding of the candidate compound with the target and ligase, and 2) hence the relationship between the drug-target binding and its cellular efficacy than the conventional occupancy-based inhibitors. This manuscript attempted to demonstrate that some relationship could be built between the SPR measurables and the cellular degradation efficacy of PROTACs, as well as a correlation between a computed property, buried surface area, with the ternary binding affinity. Indeed, the topic of the manuscript is of interest and importance to the field. However, the results provided in the current manuscript might not be strong enough to support its statement.

Major questions:

1) Given a dozen of VHL-based degraders and half-dozen of the BRD4 degraders, it is hardly to conclude a solid predictive framework for PROTAC design, which is the key selling point of the manuscript. In addition, with the limited number of compounds, the linkers of VHL degraders are more rigid, while they are more flexible ones for BRD4 degraders, make it even harder to reach a general conclusion. Also, except the linker length and rigidity, the position on the two "warheads" that linkers attach to, as well as the modifications of the two warheads are all importance factors for binding and degradation. Yet the manuscript had barely studied.

2) The authors used the calculated buried surface area (BSA) from modeled ternary structures to demonstrate the correlation between BSA and $\log(K_{LPT})$. To validate this correlation, the authors should demonstrate the accuracy of their ternary structure prediction. That is, if the ternary structure was not correct, the calculated BSA and hence the correlation would be meaningless.

3) All the affinity data and degradation data lack of error range, and the number of measurements was also not found.

4) On page 18, the authors stated that "Our work demonstrates that ternary complex affinity and cooperativity both drive the cellular activity of degraders". Given the data in Table 1, it might be fair to say that cooperativity drove the cellular activity, but not the ternary complex affinity. For instance, compound 1, 2, 7, 9 had K_{LPT} of 4.7, 33, 80 and 77 nM, respectively, yet their degradation AUC were of 13, 28, 122 and 85, which clearly showed that ternary affinity hinders rather than drives the degradation.

5) The D_{max} of the compounds are generally very small, < 10% for 10 out of the total 16 compounds. This also raises a concern for the generality of this study.

Reviewer #4:

Remarks to the Author:

This paper explores binary and ternary complex formation with PROTACs, a target protein, and a ligase that ubiquitinates the target protein. How the binary and ternary equilibrium dissociation constants and cooperativity work together to cause ubiquitination is also examined and leads to potential structure-activity relationships for the design of novel PROTACs. The experiments in the paper are sound and potential new insights are found in what factors determine the optimal ternary complex formation, which in turn leads to enhanced target protein degradation. SPR is used appropriately in the paper to interrogate how binary and ternary affinity changes affect complex stability. Appropriate cellular assays are used to examine target protein degradation as it relates to the binding affinities parsed out in the dissociation constant equilibrium studies.

I do have some questions and comments for the authors:

1. On page 7, it might clearer to show the dissociation reaction since the binding constant shown is the equilibrium dissociation constant not the equilibrium association reaction. Alternatively, the reaction arrows could be labelled with the correct equilibrium constants
 2. In the appendix, on page 36, I would like the authors to explain in more detail why $[P]=\sqrt{(KLP*KTP)}$ (eq. A16) when $d[LPT]/d[P]=0$ where LPT is at a maximum.
 3. Also in the appendix, concerning equation A18, I would like a better explanation for why the first two terms to a first approximation can be ignored when $T_t \gg T_l$. I realize the first two terms will be negligibly small when T_t is divided into those terms if T_t is much larger than T_l , but it is not obvious to me why the fourth term containing $\alpha, KLP, KTP, LPT_{max}$ does not become small too. More explanation here would help.
 4. Line 259 on page 13, should have $R^2=0.98$ and not 0.97 to be consistent with the corresponding figure.
 5. The paragraphs (lines 296-309) on page 15 confused me. Are the authors stating that for SMARCA2 degradation the ternary complex stability did not correlate with degradation, but in the BRD4 system, ternary complex stability did correlate with degradation as previously published by Roy et al.? I assume this is the case, but I think it might be worth stating explicitly in the text that different protein targets will require different biophysical properties to have optimal degradation and potency, or something to that effect. It might help the reader grasp better the point the authors are making with these paragraphs.
 6. In Figure 4B, it appears PROTAC 6 has a faster degradation rate than PROTAC 1, yet PROTAC 6 only has an $\alpha=2$ versus $\alpha=12.8$ for PROTAC 1. Is this the noise in the assay or do the authors have some speculation of why this is? I am simply curious considering the correlation for the initial degradation rate for the BRD4 system in Fig. 5 is much clearer and more highly correlated.
 7. In the caption for Figure 7, in line 536 the verb after SMARCA2 should be "are."
 8. In general, the manuscript is very well written. However, I did not see an abbreviation list and the manuscript in places is very jargon heavy with abbreviations that are not obvious to scientists not in the PROTAC field. I suggest the authors define all abbreviations used in the paper.
- I also noticed some of the sentences read awkwardly because they were missing indefinite and definite articles in the front of some nouns.

RESPONSE to REVIEWER COMMENTS:

Reviewer #1 (Remarks to the Author):

The manuscript by A. Vaish and co-worker on “Affinity and Cooperativity Modulate Ternary Complex Formation to Drive Targeted Protein Degradation” presented experimental results and analyses that supported the correlation between Protac ternary complex affinity/cooperativity and the protein degradation efficiency through the studies of two different VHL-based Protac series. The manuscript was logical developed and well written that can be published as it is. The results and conclusions were consistent with known published work primarily by A. Ciulli’s lab (references 26 and 27). However, they did not provide new insights into the relationship between the ternary complex formation and degradation. Even the “total buried surface” is not a new concept to characterize ternary complexes (for example, R. P. Law et al. *Angew. Chem. Int. Ed.* 60, 23327-23334 (2021)). The authors’ work lacks the originality and impact to the field to be qualified for publication in *Nature Communications*. To further advance the knowledge for Protac design, the studies of non-cooperative ternary complex systems that lead to degradation would be more interesting (for example, M. F. Calabrese et al. *Nat. Chem. Biol.* 17, 152-160 (2021)).

We thank the reviewer for their comments. We understand the reviewer’s perspective that this work “*did not provide new insights into the relationship between the ternary complex formation and degradation. Even the “total buried surface” is not a new concept to characterize ternary complexes (for example, R. P. Law et al. Angew. Chem. Int. Ed. 60, 23327-23334 (2021))*” and would like to provide additional information to underscore the novelty of this manuscript. The Ciulli lab (ref 26 & 27) has provided phenomenal insight into ternary complex formation and degradation using structure-based molecular design for the prototypical target BRD4. They resolved the ternary complex crystal structure with PROTAC MZ1 (ref 26), and demonstrated high cooperativity drives target degradation. In contrast to previous work, the motivation behind this manuscript was to develop a comprehensive mathematical framework to understand ternary complex attributes (i.e., binding affinity, cooperativity) and its relationship with the degradation activity (i.e., degradation rate and DC50). Additionally, we calculated total buried surface area (BSA) of the ternary complex protein-protein interface from computational modeling and correlated it with the experimental ternary complex binding affinity to analyze the relationship between these two parameters. This methodology is particularly useful for PROTAC ranking and medchem optimization efforts when no or minimal structural information is available. For

example, in this manuscript we analyzed VHL-based PROTACs for SMARCA2 with limited structural information.

We agree with the reviewer that “total buried surface area” is not a new concept, especially in the realm of protein-protein interaction prediction. This is usually one of the first metrics one computes when analyzing protein-protein complex structures. However, its significance in PROTAC induced protein-protein interactions is rarely discussed. In the *R. P. Law et al.* (Angew. Chem. Int. Ed. 60, 23327-23334 (2021)) paper as suggested by the reviewer, BSA was calculated from the solved ternary structure of FAK-GSK215-VBC. We acknowledged the work of R.P. Law et al. and cited it in the manuscript.

For further clarity and to increase our and the readers confidence in the methodologies we presented, the revised manuscript contains crystal structures of ternary complex (Figure 7C) involving a benchmark molecule [Ciulli PROTAC2 (compound **11**)] and compound **1** (with VBC/SMARCA4). Additionally, we compared top computer models with published crystal structures for three different PROTAC systems (*Supplementary Information Figure S3*) including an *in-house* ternary crystal structure (Figure 7C) of compound **11**, to verify the modeled ternary structure for BSA calculation. We have demonstrated that when structural information is limited, BSA may offer insights in ranking PROTACs regarding their ability to form ternary structures with VBC/SMARCA2.

Finally, we appreciate reviewer’s comment on discussing non-cooperative ternary complex systems. Our focus in this manuscript was to analyze cooperativity driven VHL-based PROTACs for two different targets (SMARCA2 and BRD4) and to develop a predictive framework. We have added text in the discussion section of the manuscript that future work will be focused on understanding the ternary complex attributes of non-cooperative ligases and its relationship to target degradation.

Reviewer #2 (Remarks to the Author):

This is an excellent paper by a strong team at Amgen, studying PROTAC ternary complexes formation, affinity, cooperativity and kinetics stability and attempting to correlate parameters to the Kinetics of protein degradation. It does so by utilizing the well-characterized systems of Brd4-VHL (MZ1) and SMARCA2-VHL (ACBI1) previously reported and structurally and biophysically described by our Laboratory.

The authors develop a clever mathematical model for considering the equilibria in the SPR binding assay previously described (Roy et al. 2019). The parameter beta, reporting on the relative binary binding affinities is useful and relevant. They then characterise a library of 11 SMARCA2 PROTACs (including two benchmark literature compounds, compound 6 (AU-15330) and compound 11 (ACBI1)) and 6 BET PROTACs (including compound 17 ie. MZ1 and five triazole containing compounds, following their previous work on click-chemistry library generation (Wurz et al. 2018).

The main finding of the paper is that the authors find that, broadly at least for the SMARCA2 degraders, degradation potency correlated with ternary complex affinity, while the initial rate of degradation correlated more with cooperativity. For the BRd4 degraders, they find the same as above, however (consistent with previous finding in Roy et al.) they in this case also saw that the initial rate of degradation correlated with the ternary complex half life ($t_{1/2}$) measured by SPR (although they do not plot this).

Finally, they model the SMARCA2 PROTAC ternary complex via MD simulations and find that the buried surface area (BSA) correlates with ternary K_d .

I am in two minds about this paper. On the one hand, it builds on extensive prior literature and in the most validates and corroborates previous findings that are now well established (I may have a biased view here, since this is what my laboratory has discovered and what have been the key messages of my lectures for several years now). I also feel that some of the correlations claimed are rather stretched given the poor correlation (R^2 0.6-0.7) and the low relative power given the relatively few data points.

On the other hand, the work is for the most quite solid, and it contains areas of innovation and novelty - mainly, the mathematical treatment of the SPR equilibria, and the correlative analysis of the nicely quantitative measurements carried out. It also contains nicely designed chemical series of PROTAC compounds, including appropriate benchmark literature compounds, and it addresses important aspect of understanding and predicting SAR in PROTAC mode of action, that is a critical aspect of great interest in current degrader drug design campaigns. The paper is also well written, well illustrated and well referenced.

Overall, I would be supportive of publication in Nat Commun. should a revised version of the paper satisfactorily address the below major comments and concerns, that are aimed at constructively critique the work, and for the authors consideration with a goal to improve its

scholarly contribution and placing in the context of the field.

We are grateful to the reviewer for their thorough examination of the manuscript. These comments helped us to improve the quality of this work significantly.

1. Lines 253-255:

“17 (MZ1) exhibited a faster ternary complex dissociation with BRD4BD1 compared to BRD4BD2 (Figure 5C), and ternary complex binding affinity with BRD4BD1 is 10-fold weaker compared to BRD4BD2 (Table 1).” These findings are entirely in agreement with those published by us previously (Gadd et al. 2017; Roy et al. 2019; Klein J Med Chem 2021 <https://doi.org/10.1021/acs.jmedchem.1c01496>) which warrant due references and citations here.

We thank reviewer for pointing us toward these references and they have been included in the revised manuscript.

2. Same for lines 290-291:

“reconfirming that BRD4BD2 engagement is the key driver of BRD4 degradation.” This needs citations to the above references, as per comment above

Once again, we thank reviewer for pointing these references for lines 290-291 and they have been included in the revised version of the manuscript.

3. Wrt to correlating BSA to ternary complex stability - this has been discussed and scholarly treated before [Kozicka & Thoma, Cell Chem Biol 2021, <https://doi.org/10.1016/j.chembiol.2021.04.009>; and Cowan and Ciulli, Annu Rev Biochem, <https://doi.org/10.1146/annurev-biochem-032620-104421>] and hence the authors might want to duly reference these articles.

We thank reviewer for suggesting these references and they have been included in the revised version of the manuscript in the Results section discussing BSA and ternary complex binding affinity.

4. Fig. 2A and 2D seem to have overlooked / ignored the hook effect at high PROTAC concentration? The hook effect with the PROTACs studied in this work is clearly experimentally observed in the data shown in Fig. 4A, for example. This should be clarified and discussed

Many thanks for pointing out this omission. We have added text in the Discussion section to call out the frequently observed hook effect, provided a brief description of the underlying mechanism and relationship to the observed activities of bifunctional molecules. The following text has been added or modified.

“Ternary complex formation and induced protein degradation can be inhibited by excess PROTAC leading to the frequently observed “hook effect”. The observed squelching of activity at high concentrations of bispecific molecules is caused by the unproductive binary interaction of the PROTAC with substrate or ubiquitin ligase rather than a productive ternary complex of both target and ligase. It has been previously observed that PROTACs that promote ternary complexes from cooperative protein-protein interactions suffer less from the hook effect 47, 48. In this study, we similarly observe reduced “hook effects” for those molecules that promote the greatest cooperativity Figure 4A (i.e., 1, 4, 5, and 9).”

5. Figure 3A (should this not a Table instead?) seem to report Dmax as % of remaining protein after degrader treatment. This is conventionally reported as % of degraded (depleted) protein instead please revise

We thank reviewer for their suggestions. We have changed Figure 3A into Table 1 for SMARCA2-VHL PROTACs and Table 2 for BRD4-VHL PROTACs and changed Dmax % as degraded target in the Tables.

6. Please show the chemical structure of compound 11 in a main text figure / table, and specify that it is aka ACB11. Equally, please specify that compound 6 is aka AU-15330

Once again, we thank reviewer for finding the missing structure in the main text. We have included chemical structure of compound 11 (PROTAC 2) in the main text and Table 1.

Additionally, we specify that compound 6 is AU-15330 in the main text as well as Table 1 and

Table 3. We want to clarify that compound 11 is PROTAC 2 instead of ACBI1 from Ciulli *et al.*, Nature Chemical Biology (2019) publication.

7. All BET PROTACs used in this study contain a triazole from the click chemistry, while MZ1 does not. So in addition to having longer linkers, they most likely also will have poorer cell permeability (a known limitation of triazole groups). This is seen as striking drop in degradation potency between MZ1 and all the other compounds (Fig. 5A), as well as striking drop in potency in cellular TE assay (Fig. 6B). This should be discussed. As the authors hint to, this is also reflective of the poor correlation ($R^2=0.72$) in Fig. 6C, as the cellular TE potencies of compounds 12-16 are all skewed to weak binding affinity likely due to poor cell permeability. The authors might want to perform the experiment again in permeabilized format (see Riching *et al.* Current Res Chem Biol 2021 <https://doi.org/10.1016/j.crchbi.2021.100009>; Imaide *et al.* Nat Chem Biol 2021 <https://doi.org/10.1038/s41589-021-00878-4>). Actually, in deeper inspection, it seems based on the last page of the Supporting Information that they have done this already and so have the data?

We thank again to the reviewer for bringing one of the important aspects of PROTAC design, the cellular permeability. We have measured permeability for both series of PROTACs using a NanoBRET cellular assay in live and permeabilized cells. All the data has been included in *Supplement Information Table S3 and Table S4*, and the probable impact of permeability on degradation activity has been included in the main text.

8. Table 1: please explain how AUC was calculated from the degradation assay data

We have added text to the methods section found in the supplemental information to describe our calculation of AUC. The following text was added,

“To calculate area under the curve, all points on the curve were connected and vertical lines were drawn from each point down to $y=0$, generating a series of trapezoids. The area of each trapezoid was determined using the equation $A = (1/2)(h)(b1 + b2)$, where $h = 1$ (the distance between points), $b1 = \text{POC of the first point}$ and $b2 = \text{POC of the second point}$. The sum of all trapezoids provides an approximate area under the curve (AUC).”

9. Figs 4B and 5B. For the time course experiments, time-course profiles and hence $t_{1/2}$ will be dependent on PROTAC concentration. The authors should explain the criteria followed to select the PROTAC concentration used in these experiments

We thank reviewer for catching this omission. To address this concern, we added text that describes how we picked the “doses” for time course experiments.

“To measure the rate of protein degradation, we treated A375 cells with a single concentration of each PROTAC and measured the decrease in target abundance over time, Figure 4B. The concentration chosen for each PROTAC was at or near the concentration where maximal activity was observed in our dose response assays.”

10. Figure 4E/F and 5E/F: what would the correlation plot look like if the authors plotted $\text{Log}(\text{DC}_{50})$ vs $\text{log}(\text{K ternary})$? What would the correlation plot look like if they plotted degradation potency vs $\text{log}(\alpha)$, and conversely degradation rate vs $\text{log}(\text{K ternary})$? For sake of fair comparison, and to allowed the reader to best judge, the authors should show those missing correlation plots too.

We thank reviewer for this insightful suggestion. We have included all the plots in the revised manuscript in Figure 4, Figure 5 and Figure S1 and modified the text accordingly.

11. Fig.s 4F and 5F: what would the correlation plot look like if they plotted degradation rate vs ternary complex SPR $t_{1/2}$? Based on their results discussion, a good correlation is seen for this relationship too, so the actual plot data should be shown to support the conclusion.

We thank reviewer again for this suggestion. We have included plots of degradation rate vs. ternary complex half-life in SPR in the revised manuscript in Figures 4H/5H and discussed in the text accordingly. As shown in the plot below, SMARCA2 degradation rate shows poor correlation with the ternary complex half-life. In contrast, BRD4 degradation rate shows excellent correlation with the ternary complex half-life.

12. The differential correlative trends observed between SMARCA2 and Brd4 degraders are interesting, and the authors attempt to speculate on why that might be. One main difference between these two class of PROTACs is that the BET ligand has much higher binding affinity for the Brd4 bromodomain (K_d 10-100nM) than the SMARCA2 ligand which has K_d of 200-2000 nM) for the SMARCA2 bromodomain. The authors only measure binding affinities to the VHL ligase (K_{LP}) but do not measure the K_d for the bromodomains (K_{TP}) - and they might wish to include such measurements. I am wondering if these major difference could play a role in the observed differences in correlative trends?

This is an excellent point raised by the reviewer. We included the PROTAC/target binary binding affinity (K_{TP}) in the Table S1 of Supplementary Information. As alluded by the reviewer, the SMARCA2 PROTACs showed weaker and broader K_{TP} (binding affinity range 14 -800 nM) compared to BRD4 PROTACs (binding affinity range 10-25 nM). This difference in the binary interaction could be attributed to the poor correlation in the SMARCA2 plots compared to BRD4.

I agree to waive anonymity in a spirit to enhance the transparency of the peer review process, Alessio Ciulli

Reviewer #3 (Remarks to the Author):

Unlike the conventional inhibitors, the pharmacology of PROTACs involve additional steps to take effect, i.e. the formation of the ternary structure, the recruitment of ubiquitin and hence the degradation, in addition to the binding of the inhibitor and target. Therefore, this makes it more difficult to 1) define the binding of the candidate compound with the target and ligase, and 2) hence the relationship between the drug-target binding and its cellular efficacy than the conventional occupancy-based inhibitors. This manuscript attempted to demonstrate that some relationship could be built between the SPR measurables and the cellular degradation efficacy of PROTACs, as well as a correlation between a computed property, buried surface area, with the ternary binding affinity. Indeed, the topic of the manuscript is of interest and importance to the field. However, the results provided in the current manuscript might not be strong enough to support its statement.

Major questions:

1) Given a dozen of VHL-based degraders and half-dozen of the BRD4 degraders, it is hardly to conclude a solid predictive framework for PROTAC design, which is the key selling point of the manuscript. In addition, with the limited number of compounds, the linkers of VHL degraders are more rigid, while they are more flexible ones for BRD4 degraders, make it even harder to reach a general conclusion. Also, except the linker length and rigidity, the position on the two “warheads” that linkers attach to, as well as the modifications of the two warheads are all importance factors for binding and degradation. Yet the manuscript had barely studied.

We agree with the reviewer that the compound series we generated are not staggeringly large. We synthesized compounds that encoded what we hoped would represent significant differences in linker length and ligand attachment chemistry. When the activity and SPR attributes of these compounds were captured, we realized that we had found meaningful correlations and thus considered the chemistry effort sufficient. In our revision of the manuscript, we have added additional plots and provided two crystal structures to better capture and improve upon the significance of our findings. Though we would have liked to have added additional compounds to each series, we could not do so for the revision of this manuscript.

2) The authors used the calculated buried surface area (BSA) from modeled ternary structures to demonstrate the correlation between BSA and $\log(K_{LPT})$. To validate this correlation, the authors should demonstrate the accuracy of their ternary structure prediction. That is, if the ternary structure was not correct, the calculated BSA and hence the correlation would be meaningless.

We thank reviewer for calling out this important point. We have included three major changes in the revised manuscript which gave us confidence about the modeling workflow implemented as follows:

- i) We have tested the modeling procedure on PROTAC induced complexes with published crystal structures including VHL-MZ1-BRD4, VHL-PROTAC2-SMARCA2, and VHL-PROTAC6-BclxL (*Supplementary Information Figure S3*). We have also included *in-house* crystal structure of VHL-Cmpd **11**-SMARCA2 in the model verification (Figure 7C). The near-native model is always among the top 3 poses. Additionally, we were able to get a moderate resolution crystal structure of compound **1** with SMARCA4/VHL (Figure 7C). We demonstrated that for compound **1**, even though the top models generated using the ternary structure prediction workflow look different than the crystal structure of SMARCA4-Cmpd **1**-VHL, the model is well within the structure ensemble from MD simulations starting with SMARCA2 substituted for SMARCA4 (*Supplementary Information Figure S4*).
- ii) We compared calculated BSA for compound **11** from modeling workflow to that from the published and *in house* crystal structures, and they are in a good agreement. Details are provided in the *Supplementary Information Table S2*.

BSA of compound 11 from Modeling Workflow	1268 Å ²
BSA of compound 11 from in-house crystal structure	1256 Å ²
BSA of compound 11 from published crystal structure (6HAX)	1266 Å ²

- iii) We have included the error bars in Figure 7B to reflect the top poses of the ternary complexes in the scatterplot for correlation analysis.

We added Figures S3 and S4 and the text provided below in the supplementary information.

*“To verify the modeled ternary structure, we compared top models with published crystal structures for three different PROTAC systems including a in house ternary crystal structure of compound **11**. Supplementary Figure S3 shows the comparison between the top models and the*

crystal structures. Additionally, the BSA of compound **11** from modeled complex is consistent with that calculated from in-house (Figure 7C) or published crystal structures. The internal attempt to solve the ternary crystal structure between VBC and SMARCA2BD with compound **1** was not successful but we obtained a ternary crystal structure with the paralog of SMARCA2, SMARCA4 (Figure 7C). We modeled the SMARCA2 ternary structure with compound **1** using this crystal structure by substituting SMARCA4 with SMARCA2 and performed MD simulations to relax the system. The SMARCA2BD-cmpd **1**-VBC complex appears to be more flexible than its SMARCA4 counterpart. The three top models of compound **1** induced ternary complex from the modeling workflow are all well within the structural ensemble generated from simulating the crystal structure derived SMARCA2BD-cmpd **1**-VBC system (Supplementary Figure S4).”

Figure S3: Comparison of the top model (slate/yellow/green) and the crystal structures (light grey) in three known PROTAC induced complexes with target warhead RMSD shown on the bottom.

Figure S4: Comparison of the MD simulation snapshots for (A) the SMARCA2^{BD}-Cmpd 1-VBC model and (B) the SMARCA4^{BD}-Cmpd 1-VBC crystal structure. The top two models (*rainbow*) from the ternary structure modeling workflow are well within the structural ensemble from MD simulations of SMARCA2-cmpd 1-VHL. The structures are aligned using VHL.

3) All the affinity data and degradation data lack of error range, and the number of measurements was also not found.

We thank reviewer for pointing this out. We have included error range and the number of measurements in Table 3.

“*Error is SEMs for N=3, MSD curves for DC50/AUC: Curves are a best fit of means from N = 3 biologically independent experiments, +/- SEM; Time course curves for degradation rate: Curves are a best fit (one-phase decay) of means from N = 3 biologically independent experiments, +/- SEM.”

4) On page 18, the authors stated that “Our work demonstrates that ternary complex affinity and cooperativity both drive the cellular activity of degraders”. Given the data in Table 1, it might be fair to say that cooperativity drove the cellular activity, but not the ternary complex affinity. For instance, compound 1, 2, 7, 9 had K_{LPT} of 4.7, 33, 80 and 77 nM, respectively, yet their degradation AUC were of 13, 28, 122 and 85, which clearly showed that ternary affinity hinders rather than drives the degradation.

We have added scatterplots that show more clearly the correlations of affinity, cooperativity, and complex half-life to DC50, AUC, and initial rate (Figure 4 and Figure 5). In general, a complex with greater affinity, lower KLPT, results in a lower AUC, reflective of less SMARCA2 or BRD4 remaining within the timeframe of the assay. We hope that the improvements to the figures help to clarify these relationships.

5) The D_{max} of the compounds are generally very small, < 10% for 10 out of the total 16 compounds. This also raises a concern for the generality of this study.

We thank reviewer for the careful reading of the manuscript, and the insightful comments which helped us to improve this work. In the revised manuscript we have changed Dmax % as degraded target in the Table 1 and 3, in line with the way it's reported in the PROTAC publications. In this study, we have included molecules with degradation activities ranging from highly potent degraders [i.e., compound 1: DC50 = 28 nM and Dmax(%)= 98] to medium potency [i.e., compound 11: DC50 = 221 nM and Dmax(%)= 53] and inactive degraders [i.e., compound 3: DC50 > 10000 nM]. This allowed us to measure distribution of degradation activity (i.e., AUC and DC50) for SMARCA2 and BRD4 degraders for studying correlation between degradation activity and ternary complex formation attributes.

Reviewer #4 (Remarks to the Author):

This paper explores binary and ternary complex formation with PROTACs, a target protein, and a ligase that ubiquitinates the target protein. How the binary and ternary equilibrium dissociation constants and cooperativity work together to cause ubiquitination is also examined and leads to potential structure-activity relationships for the design of novel PROTACs. The experiments in the paper are sound and potential new insights are found in what factors determine the optimal ternary complex formation, which in turn leads to enhanced target protein degradation. SPR is used appropriately in the paper to interrogate how binary and ternary affinity changes affect complex stability. Appropriate cellular assays are used to examine target protein degradation as it relates to the binding affinities parsed out in the dissociation constant equilibrium studies.

I do have some questions and comments for the authors:

We sincerely thank reviewer for underscoring the key findings of our manuscript. All the questions/comments are addressed below.

1. On page 7, it might clearer to show the dissociation reaction since the binding constant shown is the equilibrium dissociation constant not the equilibrium association reaction. Alternatively, the reaction arrows could be labelled with the correct equilibrium constants

We thank reviewer for this suggestion. We have changed the direction of the arrow in the revised manuscript.

2. In the appendix, on page 36, I would like the authors to explain in more detail why $[P]=\sqrt{KLP \cdot KTP}$ (eq. A16) when $d[LPT]/d[P]=0$ where LPT is at a maximum.

We have added additional steps A17 to A19 to explain how we got $[P]=\sqrt{KLP \cdot KTP}$ in the Appendix.

3. Also in the appendix, concerning equation A18, I would like a better explanation for why the first two terms to a first approximation can be ignored when $T_t \gg L_t$. I realize the first two terms will be negligibly small when T_t is divided into those terms if T_t is much larger than L_t , but it is not obvious to me why the fourth term containing α , KLP , KTP , LPT_{max} does not become small too. More explanation here would help.

We thank reviewer again for pointing this out. We have modified the explanation for clarity in the Appendix. In A21, the first two terms are very small and approximately equal and opposite, thus, they can cancel each other out. The remaining terms are rearranged to transform into A22.

We expanded eq A20:

$$[LPT]_{max}^2 - [LPT]_{max} \left([L]_t + [T]_t + \frac{(\sqrt{K_{LP}} + \sqrt{K_{TP}})^2}{\alpha} \right) + [L]_t [T]_t = 0 \quad (A20)$$

Followed by dividing it by $[T]_t$ as follows:

$$[LPT]_{max}^2/[T]_t - [LPT]_{max}[L]_t/[T]_t - [LPT]_{max} \left(1 + \frac{(\sqrt{K_{LP}} + \sqrt{K_{TP}})^2}{\alpha[T]_t} \right) + [L]_t = 0 \quad (A21)$$

As a first order approximation, the first and second terms of eq A21 ($[T]_t \gg [L]_t$ or $[LPT]_{max}$), can cancel each other out, which transforms it into:

$$\frac{[LPT]_{max}}{[L]_t} \cong \frac{\alpha}{\left(\alpha + \frac{(\sqrt{K_{LP}} + \sqrt{K_{TP}})^2}{[T]_t} \right)} \quad (A22)$$

4. Line 259 on page 13, should have R2=0.98 and not 0.97 to be consistent with the corresponding figure.

We thank reviewer for the suggestion. We added several new correlation plots in the revised manuscript for a comprehensive analysis of different parameters discussed in this work. We used Pearson correlation coefficient (r) for correlation analysis and provided the statistical significance of correlation in the revised manuscript.

5. The paragraphs (lines 296-309) on page 15 confused me. Are the authors stating that for SMARCA2 degradation the ternary complex stability did not correlate with degradation, but in the BRD4 system, ternary complex stability did correlate with degradation as previously published by Roy et al.? I assume this is the case, but I think it might be worth stating explicitly in the text that different protein targets will require different biophysical properties to have optimal degradation and potency, or something to that effect. It might help the reader grasp better the point the authors are making with these paragraphs.

Once again, we thank reviewer for pointing this out. We have added, plots 4H and 5H in the revised manuscript to underscore that for SMARCA2 degraders there is a poor correlation between degradation rate and ternary complex stability compared to the excellent correlation between degradation rate and ternary complex stability for BRD4 degraders. Additionally, we have slightly edited the main text to clarify this point further.

“We can only speculate that the requirement for complex stability will vary between different substrate proteins and possibly between series of PROTACs for the same substrate and that for our most active SMARCA2 degraders, we have surpassed a threshold of induced ternary complex stability and any increases in stability no longer drive more efficient ubiquitination and degradation.”

6. In Figure 4B, it appears PROTAC 6 has a faster degradation rate than PROTAC 1, yet PROTAC 6 only has an $\alpha=2$ versus $\alpha=12.8$ for PROTAC 1. Is this the noise in the assay or do the authors have some speculation of why this is? I am simply curious considering the correlation for the initial degradation rate for the BRD4 system in Fig. 5 is much clearer and more highly correlated.

Thank you for your careful examination of the data. We have addressed the deviation of compound 6 from the trend of the series by describing how cooperativity is calculated and pointing out the relatively high affinity that compound 6 exhibits towards VHL alone. This high affinity interaction with VHL suppresses the calculated cooperativity despite the high ternary complex affinity measured. The following text was added or edited in the main text.

“Considering the entire data set, initial SMARCA2 degradation rates demonstrated a negative correlation with ternary complex binding affinity ($r = -0.7$) and a positive correlation with cooperativity ($r = .67$) (Figure 4 E and F). Importantly, this work illustrates the need to track both ternary complex affinity and cooperativity. Though compound 6 (AU-15330) promotes high ternary complex affinity, it also exhibits relatively modest cooperativity ($\alpha = 2$). Cooperativity is calculated by taking the ratio of a PROTAC’s binary to ternary complex binding affinities. Because

compound 6 (AU-15330) has shown higher binary binding affinity, PROTAC to VHL (KLP = 11 nM), relative to other PROTACs in the series (Table1), the calculated cooperativity is also lowered relative to PROTACs with similar affinity, i.e., compound 1. Despite these unique deviations from the trend, the SMARCA2 PROTAC series shows that positive cooperativity promotes higher ternary complex affinity, higher potency, and higher rates of target degradation."

7. In the caption for Figure 7, in line 536 the verb after SMARCA2 should be "are."

We thank reviewer for identifying this mistake. We fixed it in the revised manuscript.

8. In general, the manuscript is very well written. However, I did not see an abbreviation list and the manuscript in places is very jargon heavy with abbreviations that are not obvious to scientists not in the PROTAC field. I suggest the authors define all abbreviations used in the paper. I also noticed some of the sentences read awkwardly because they were missing indefinite and definite articles in the front of some nouns.

We sincerely thank reviewer for their feedback on our work and insightful suggestions. We have added the abbreviation list in the revised manuscript and modified the text wherever it seems awkward.

REVIEWER COMMENTS

Reviewer #2 (Remarks to the Author):

The authors have done an excellent job at addressing my comments and critique at first review, as well as that of the other referees. The analysis of the correlative trends from the data is more comprehensive now and more thoroughly discussed also in place of the prior literature. Also the new data included at revision, particularly the cellular target engagement data, clearly shows a major differentiation between the two series, establishing the BRD4 PROTAC series as more permeability limited and hence more dependent on ternary complex formation/dissociation kinetic and thermodynamic parameters. I am satisfied with the revised version and I can now recommend publication of the article, should this meet with my fellow Editors and REviewers support too.

I have only one point for the attention of the authors. In their new sentence at the end of the manuscript, the authors might want to tone down their qualifying statement about CRBN and IAP PROTAC degraders not being dependent on cooperatively, which remains to be established. this statement seems to extrapolate from reports which showed some CRBN and IAP based PROTAC degraders that are not cooperative; however, just because some are shown to be like that, does not mean that all will not dependent on cooperatively. So I believe the sentence " ... particularly those which don't require positive cooperativity for target degradation (i.e., CRBN, cIAP) " should be revised and made not so qualifying.

Also, not just CRBN but also VHL have flexibility in the Cullin RING system.

Reviewer #3 (Remarks to the Author):

The authors have well responded to my questions and concerns.

Reviewer #4 (Remarks to the Author):

The authors answered all my questions satisfactorily. This paper should now be published.

Reviewer #5 (Remarks to the Author):

"Affinity and Cooperativity Modulate Ternary Complex Formation to Drive Targeted Protein Degradation"

This is a well-written and interesting manuscript by A. Vaish and co-workers describing biophysical and cellular studies on a series of VHL-based PROTACs targeting Brd4 and SMARCA2 to relate PROTAC ternary complex affinity and complex stability to parameters of protein degradation efficiency, including degradation kinetics.

Many of the methodologies employed in this study have previously been established in the literature, such as biophysical approaches to study PROTAC ternary complex formation/cooperativity and the SPR ternary complex assay, as well as comparison to ternary complex buried surface area in PROTAC structures. However, the authors build on this earlier work in a few respects. One aspect is attempting to provide a more detailed mathematical framework to relate three-body equilibria to ternary complex formation to detection using SPR. A second is in over-viewing a computational ternary complex modelling pipeline able to yield informative structures (consistent with ones determined experimentally) and relating derived BSA to degradation efficiency.

The two PROTAC systems studied in this paper represent VCB/bromodomain interactions that also have been studied in detail previously and in this sense, there is perhaps an opportunity missed to extend the analysis to another target class (eg. kinases or more flexible targets). However, there is value to the field in reproducing and corroborating earlier studies, both with existing reference compounds and in extending to a new compound series.

Focussing on the crystallography data, the study presents two crystal structures determined in house,

one of which is a new complex (SMARCA4-Compound 11-VBC, to 3.7 Å) and is a complex that has been published previously (SMARCA2-Compound 1-VBC, to 2.7 Å), but is reproduced primarily to validate the crystallography pipeline. The experimental crystal structures of the ternary complexes are used to support the overall pose for the ternary complex from computational modelling and inform buried surface area analysis. The crystallographic data collection, processing and structure determination appear appropriate, although there are some important details that should be included, outlined below.

Overall this study is well-developed and results are clearly presented and appropriately interpreted, with some important points below which should be addressed.

Major points:

1. Crystallography

1a. Only limited experimental and structural details are provided for the crystallography studies and some important information is absent. The authors should provide a table summarizing crystallographic statistics for the two in house structures they present. Additionally, the crystallization and cryoprotection conditions for the crystals used in data collection should be included in the experimental section.

1b. The SMARCA2-Compound 1-VBC complex structure has been previously published to a higher resolution than the new structure determined in this current study (pdb: 6hax, 2.3 Å), but there is no comparison of whether these structures are essentially identical or differ in material respects. In Figure 7C the authors should instead provide an overlay or side-by-side comparison of the existing and new structures (eg. calculate main chain RMSD value between the two structures) and or differences noted in experimental section (crystallization conditions, space group/unit cell). The higher resolution structure would seem to be the more relevant primary comparator for the computational models, including in Figure S4, or differences should be discussed. At present buried surface area analysis is the only direct comparison made.

1c. The authors should include Supporting Figures that illustrating the electron density/fitting for the two structures, in particular SMARCA4-Compound 11-VBC. Eg. in Figure S6 include a panel of the protein modelled into in the electron density 2Fobs-Fcalc (2Fo-Fc) for the protein/ligand/protein interface showing side-chains, and an unbiased omit map (Fo-Fc) for the modelled ligand. It should be noted that the structures appear to be modelled appropriately and electron density from both structures appears to support the modelled conformation, however this should be illustrated for the reader.

2. Other sections

2a. SPR-derived ternary KD values and off-rate/half-life represent a key piece of data for comparisons made throughout the paper (eg. to degradation kinetics and buried surface area), however no fitted SPR sensorgrams are currently included in the paper to enable an evaluation of goodness of SPR fitting used to derive KD values or kinetics. The authors should include in supplementary data a figure with a representative sensorgram for each protein/compound combination tested that shows curve fitting overlaid. Fitted kinetic parameters (on rate, off rate, KD, complex half life) and uncertainties/errors in these values should be included for the data summarised in Table 3 and Figure 4G, H. Additionally the experimental section should include a description of the method of fitting to derive KD values.

Table S1 – Errors for KTP and number of independent experiments for SPR data should be included.

Minor points:

- The refinement statistics and model validation for the SMARCA2-Compound 1-VBC structure are less favourable, which is to be expected for a lower resolution structure, but may suggest this model would benefit from further refinement. In particular the clash score for the SMARCA2-Compound 1-VBC complex appears relatively high (a number of residues with > 1 Å overlap). This does not affect conclusions relating to the overall ternary complex pose, however authors may wish to address these in the deposited PDB to provide the best possible reference structure for follow-on studies.

- Line 87-90: "PROTAC-mediated ternary complex formation has been analyzed by proximity-based assays, in which one of the labeled-proteins is titrated against the other two components to generate a dose-response (DR) curve ... The salient characteristic of these binding interactions is a bell-shaped curve". This description seems to be inaccurate as more typically the concentrations of each labelled target proteins would be kept constant and the PROTAC molecule would be titrated in dose-response, leading to a characteristic bell-shaped curve - in which, at low PROTAC concentrations insufficient bivalent exists to form the maximum achievable level of ternary complex, and at high concentrations binary interaction of PROTAC with either target protein begin to dominate, leading to a drop in ternary complex achieved. Titrating one labelled target protein to high concentration with a constant concentration of PROTAC would not typically cause such a hook effect, rather simply saturate at one end. Suggest amending this to refer to *titration of bivalent molecule/PROTAC*.

Line 93: "these methods preclude" – they do not preclude, but experiments can be designed in such a way as to avoid, the hook effect.

Line 96: "ternary complex formation can be potently influenced by underlying protein-protein interactions (PPIs)." This is a valid statement, but in addition to protein-protein interactions also protein-ligand and -linker interactions for bivalent molecules have been shown to contribute to the interface of the complex (Eg. MZ1/pdb:5t35), it is suggested to broaden this statement to include these possibilities.

Line 161: The authors do not explain clearly why "In the SPR assay, $[T]_t \approx 25 \cdot K_{TP}$ for maintaining the binary complex". Maintaining the target concentration at $\sim 25 \times K_D$ for the target/PROTAC interaction seems reasonable (albeit arbitrary), but the rationale for this is not clearly outlined in the text. Perhaps relating this to a concept such as fractional ligand occupancy may assist readers to implement this in their own work. In practice $\sim 25 \cdot K_{TP}$ has also not experimentally been used for all compounds evaluated in this study – eg. for compound 10 ($K_{TP} \sim 865$ nM) using a bromodomain target concentration of 5 μ M actually represents $[T]_t \sim 6 \times K_{TP}$.

Line 273: "Richling et al." should be "Riching et al."

Line 412: "under saturating conditions" does not specify what is saturating

SI Lines 106 and 1109 "Availability Index" and "Permeability Index" – authors may wish to make these titles consistent

SI Line 1032 "resulted into average diffraction with 3.7 Å resolution" – suggest "resulted in diffraction with average 3.7 Å resolution"

RESPONSE to REVIEWER COMMENTS:

Reviewer #2 (Remarks to the Author):

The authors have done an excellent job at addressing my comments and critique at first review, as well as that of the other referees. The analysis of the correlative trends from the data is more comprehensive now and more thoroughly discussed also in place of the prior literature. Also the new data included at revision, particularly the cellular target engagement data, clearly shows a major differentiation between the two series, establishing the BRD4 PROTAC series as more permeability limited and hence more dependent on ternary complex formation/dissociation kinetic and thermodynamic parameters. I am satisfied with the revised version and I can now recommend publication of the article, should this meet with my fellow Editors and Reviewers support too.

We sincerely thank reviewer #2 for the insightful feedback in our manuscript which significantly improved its impact. We are looking forward to the publication of our findings.

I have only one point for the attention of the authors. In their new sentence at the end of the manuscript, the authors might want to tone down their qualifying statement about CRBN and IAP PROTAC degraders not being dependent on cooperatively, which remains to be established. this statement seems to extrapolate from reports which showed some CRBN and IAP based PROTAC degraders that are not cooperative; however, just because some are shown to be like that, does not mean that all will not dependent on cooperatively. So I believe the sentence “ ... particularly those which don’t require positive cooperativity for target degradation (i.e., CRBN, cIAP) ” should be revised and made not so qualifying.

Also, not just CRBN but also VHL have flexibility in the Cullin RING system.

We agree with the reviewer that “some CRBN and IAP based PROTAC degraders that are not cooperative; however, just because some are shown to be like that, does not mean that all will not dependent on cooperatively”.

We modified the sentence at the end of manuscript as:

“Extending the SPR and cell-based methods presented in this manuscript to other ligase-target pairs^{52, 53} will broaden our growing appreciation of the unique SAR that arises during the engineering of PROTACs. We hope that advances in structure-based computational modeling⁵⁴, free-energy perturbation (FEP) methods⁵⁵ and cryo-electron microscopy to probe the conformational states of ternary complexes⁵⁶, will make PROTAC design more efficient.”

Reviewer #3 (Remarks to the Author):

The authors have well responded to my questions and concerns.

We sincerely thank reviewer #3 for the insightful comments in our manuscript which significantly improved its quality and scope. We are looking forward to the publication of our findings.

Reviewer #4 (Remarks to the Author):

The authors answered all my questions satisfactorily. This paper should now be published.

We sincerely thank reviewer #4 for the insightful feedback during the review process which significantly improved its impact. We are looking forward to the publication of our manuscript.

Reviewer #5 (Remarks to the Author):

“Affinity and Cooperativity Modulate Ternary Complex Formation to Drive Targeted Protein Degradation”

This is a well-written and interesting manuscript by A. Vaish and co-workers describing biophysical and cellular studies on a series of VHL-based PROTACs targeting Brd4 and SMARCA2 to relate PROTAC ternary complex affinity and complex stability to parameters of protein degradation efficiency, including degradation kinetics.

Many of the methodologies employed in this study have previously been established in the literature, such as biophysical approaches to study PROTAC ternary complex formation/cooperativity and the SPR ternary complex assay, as well as comparison to ternary complex buried surface area in PROTAC structures. However, the authors build on this earlier work in a few respects. One aspect is attempting to provide a more detailed mathematical framework to relate three-body equilibria to ternary complex formation to detection using SPR. A second is in over-viewing a computational ternary complex modelling pipeline able to yield informative structures (consistent with ones determined experimentally) and relating derived BSA to degradation efficiency.

The two PROTAC systems studied in this paper represent VCB/bromodomain interactions that also have been studied in detail previously and in this sense, there is perhaps an opportunity missed to extend the analysis to another target class (eg. kinases or more flexible targets). However, there is value to the field in reproducing and corroborating earlier studies, both with existing reference compounds and in extending to a new compound series.

Focussing on the crystallography data, the study presents two crystal structures determined in house, one of which is a new complex (SMARCA4-Compound 11-VBC, to 3.7 Å) and is a complex that has been published previously (SMARCA2-Compound 1-VBC, to 2.7 Å), but is reproduced primarily to validate the crystallography pipeline. The experimental crystal structures of the ternary complexes are used to support the overall pose for the ternary complex from computational modelling and inform buried surface area analysis. The crystallographic data collection, processing and structure determination appear appropriate, although there are some important details that should be included, outlined below.

Overall this study is well-developed and results are clearly presented and appropriately interpreted, with some important points below which should be addressed.

Major points:

1. Crystallography

1a. Only limited experimental and structural details are provided for the crystallography studies and some important information is absent. The authors should provide a table summarizing crystallographic statistics for the two in house structures they present. Additionally, the

crystallization and cryoprotection conditions for the crystals used in data collection should be included in the experimental section.

We thank reviewer for pointing this out. In the revised manuscript all the experimental and structural details are provided for the two *in-house* crystal structures in the Supplementary Information.

“Table S5. Crystallographic data collection and refinement statistics for SMARCA2^{BD}/Cmpd11/VBC and SMARCA4^{BD}/Cmpd1/VBC reported in the manuscript.”

1b. The SMARCA2-Compound 1-VBC complex structure has been previously published to a higher resolution than the new structure determined in this current study (pdb: 6hax, 2.3 Å), but there is no comparison of whether these structures are essentially identical or differ in material respects. In Figure 7C the authors should instead provide an overlay or side-by-side comparison of the existing and new structures (eg. calculate main chain RMSD value between the two structures) and or differences noted in experimental section (crystallization conditions, space group/unit cell). The higher resolution structure would seem to be the more relevant primary comparator for the computational models, including in Figure S4, or differences should be discussed. At present buried surface area analysis is the only direct comparison made.

Once again, we thank reviewer for calling out this important point. These structures are essentially identical, crystallizes in the same space group, and having similar unit cell parameters. We have included overlay of the existing ternary complex structure (6HAX) and in-house (8G1P) in Figure 7C in the main text and provided detailed overview of both copies in the Supplementary Information. Additionally, we have performed RMSD analysis of these two structures indicating they are very similar with RMSD < 3Å.

“As illustrated in Figure 7C, overlay of in-house ternary complex crystal structure (PDB: 8G1P) of compound II (PROTAC 2) is similar to that of the published structure (PDB: 6HAX), with a RMSD less than 3Å. Additionally, the BSA of modeled complex with compound II (PROTAC 2) is consistent with that calculated from both 8G1P and 6HAX³¹ (Supplementary Information).”

Figure S13. (A) Overview of overlay of 6HAX (literature) and 8G1P (*in-house*) crystal structures with VBC-Cmpd **11**-SMARCA2^{BD}. Structures aligned using VHL (B) 6HAX copy 1 is shown in wheat and copy of 8G1P structure is shown in pale cyan, (C) 6HAX copy 2 is shown in wheat and copy of 8G1P structure is shown in purple.

	8G1P_C_G	8G1P_F_H	6HAX_C_A	6HAX_F_E
8G1P_C_G		2.66	2.61	0.98
8G1P_F_H			0.66	2.62
6HAX_C_A				2.48
6HAX_F_E				

Table S6. The pairwise C α root mean squared deviation (RMSD) in Å between all ternary complex structure copies in the SMARCA2-Cmpd**11**-VBC crystal structures [PDB IDs: 8G1P (*in-house*) and 6HAX (published)]. VBC is used in the structure alignments and the RMSD is computed for the SMARCA2 C α atoms. The names of the table entries contain the PDB ID followed by the chain IDs for VHL and for SMARCA2 of each structure copy used.

1c. The authors should include Supporting Figures that illustrating the electron density/fitting for the two structures, in particular SMARCA4-Compound 11-VBC. Eg. in Figure S6 include a panel of the protein modelled into in the electron density $2F_{obs}-F_{calc}$ ($2F_o-F_c$) for the protein/ligand/protein interface showing side-chains, and an unbiased omit map (F_o-F_c) for the modelled ligand. It should be noted that the structures appear to be modelled appropriately and electron density from both structures appears to support the modelled conformation, however this should be illustrated for the reader.

We thank reviewer for this suggestion. We have included electron density maps for the two structures in the Supplementary Information. As shown in Figure S15, omit map (F_o-F_c) of compound **1**, sidechain density at the protein/protein interface at 3.7Å resolution is resolved for some interactions but not for all the possible interactions. However, it is reported that these residues are positioned suitably for interactions.

Figure S15. Electron density maps for VBC-Cmpd **1**-SMARCA4^{BD} crystals. (A) F_o-F_c omit map (green meshes) of compound **1** contoured at 3.0σ . (B) Analysis of crystal contacts at VHL-SMARCA4 interface. (C) $2F_o-F_c$ map (blue meshes) of compound **1** contoured at 1.0σ .

Figure S12. Electron density maps for VBC-Cmpd 11-SMARCA2 crystals. (A) F_o-F_c omit map (green meshes) of Cmpd 11 in contoured at 3.0σ . (B) Analysis of crystal contacts at VHL-SMARCA2 interface. (C) $2F_o-F_c$ map (blue meshes) of compound **11** (shown in yellow sticks model) in VBC-Cmpd 11-SMARCA2 contoured at 1.0σ .

2. Other sections

2a. SPR-derived ternary KD values and off-rate/half-life represent a key piece of data for comparisons made throughout the paper (eg. to degradation kinetics and buried surface area), however no fitted SPR sensorgrams are currently included in the paper to enable an evaluation of goodness of SPR fitting used to derive KD values or kinetics. The authors should include in supplementary data a figure with a representative sensorgram for each protein/compound combination tested that shows curve fitting overlaid. Fitted kinetic parameters (on rate, off rate, KD, complex half life) and uncertainties/errors in these values should be included for the data summarised in Table 3 and Figure 4G, H. Additionally the experimental section should include a description of the method of fitting to derive KD values.

Table S1 – Errors for KTP and number of independent experiments for SPR data should be included.

We thank reviewer for catching this omission. We have included tables for both SMARC2 and BRD4 degraders in the Supplementary Information describing all the SPR-derived parameters. Additionally, we have included all the representative fitted sensorgrams along with data analysis details in the Supplementary Information.

Table S1. SPR-measured ternary complex binding parameters of SMARCA2 degraders.

Compound	k_a ($M^{-1}s^{-1}$) $\times 10^5$	\pm	k_d (s^{-1})	\pm	$t_{1/2}$ (s)	\pm	K_{LPT} (nM)	\pm	K_{LPT} (nM) (Steady-State)	\pm
1	35.2	3.9	0.017	0.009	41.5	2.4	4.7	1	5.6	0.8
2	8.7	1.7	0.031	0.002	24.3	2.1	33	4	37	5
3*	Fast		Fast		2				500	140
4	5.9	0.95	0.039	0.002	18.1	1.8	64	7	61	5.5
5	20.7	1.8	0.09	0.013	7.4	0.85	46.8	6.7	36	5
6 (AU-15330)	57.4	17.5	0.034	0.012	20.2	6.7	5.6	2	7.5	2.2
7	21.7	4.4	0.19	0.02	3.6	0.24	80	6	76	8
8	37	2.7	0.11	0.006	6.3	0.33	27	3	32	5
9*	Fast		Fast		2				77	8
10	3.5	0.76	0.08	0.007	8.4	0.73	250	30	230	60
11	11.3	2.4	0.12	0.011	5.9	0.6	108	14	121	25

SPR analysis was performed using 1:1 Langmuir interaction or steady-state affinity models. Error is SEMs for $N=3$ or SD for $N=2$. χ^2 and uniqueness (U) values were used to determine the quality of fitted parameters and confidence in the results. Steady-state binding affinity for compounds **3** and **10** was calculated using steady-state affinity model with constant R_{max} (90 RU, based on compound **1**); * ternary complex half-life of 2 s was assumed for compounds **3** and **9** with fast-off binding kinetics.

Table S3. SPR-measured ternary complex binding parameters of BRD4 degraders.

Compound	k_a ($M^{-1}s^{-1}$) $\times 10^5$ \pm	k_d (s^{-1}) \pm	$t_{1/2}$ (s) \pm	K_{LPT} (nM) \pm	K_{LPT} (nM) \pm (Steady-State)
12	66.2 12.3	0.038 0.0002	18.7 0.1	6 1	4.8 1.3
13	63.4 8.8	0.058 0.0005	12.2 0.15	10 2	7.9 2.1
14	96.4 16.2	0.25 0.02	4.4 0.64	26.5 4	25.6 2.3
15	31 10.3	0.13 0.01	5.5 0.44	39 5	35.6 5.9
16*	Fast	Fast	2		49 6
17 (MZ1)	96 22.3	0.015 0.002	54.8 5.7	1.5 0.5	1 0.3

SPR analysis was performed using 1:1 Langmuir interaction or steady-state affinity models. Error is SEMs for $N=3$ or SD for $N=2$. χ^2 and uniqueness (U) values were used to determine the quality of fitted parameters and confidence in the results; * ternary complex half-life of 2 s was assumed for compound **16** with fast-off binding kinetics.

Minor points:

- The refinement statistics and model validation for the SMARCA2-Compound 1-VBC structure are less favourable, which is to be expected for a lower resolution structure, but may suggest this model would benefit from further refinement. In particular the clash score for the SMARCA2-Compound 1-VBC complex appears relatively high (a number of residues with > 1 Å overlap). This does not affect conclusions relating to the overall ternary complex pose, however authors may wish to address these in the deposited PDB to provide the best possible reference structure for follow-on studies.

We thank reviewer for this suggestion. We have attempted further refinement of SMARCA2-Compound 1-VBC structure, but the refinement statistics reported in the Table S5 in the Supplementary Information were the best one that was obtained at 3.7 Å resolution. The clash score was addressed in the new deposition and was reduced from 25 to 16 as shown in the new validation report.

Table S5. Crystallographic data collection and refinement statistics for SMARCA2^{BD}/Cmpd11/VBC and SMARCA4^{BD}/Cmpd1/VBC reported in the manuscript.

	SMARCA2 ^{BD} : Cmpd11:VBC	SMARCA4 ^{BD} : Cmpd 1 :VBC
Data collection		
Space group	P 21 21 21	P 62
Cell dimensions		
a, b, c (Å)	79.523 115.672 119.654	158.163 158.163 44.857
α, β, γ (°)	90 90 90	90 90 120
Resolution (Å)	2.7(2.797-2.7)	3.73 (3.864 - 3.73)
R _{merge}	0.21 (1.214)	0.34 (0.989)
I / σ I	10.9 (2.4)	7.7 (2.8)
Completeness (%)	99.94 (100.00)	99.81 (99.56)
Redundancy	13.3 (13.8)	10.8 (11)
Refinement		
Resolution (Å)	2.7	3.73
No. reflections	31049	6944
R _{work} / R _{free}	20.9 / 26.9	23.4 / 32.5
No. atoms		
Protein	7266	3420
Ligand/ion	144	67
Water	193	4
B-factors		
Protein	44.17	78
Ligand/ion	37.09	55.63
Water	38.62	31.61
R.m.s. deviations		
Bond lengths (Å)	0.004	0.004
Bond angles (°)	0.85	0.72

- Line 87-90: “PROTAC-mediated ternary complex formation has been analyzed by proximity-based assays, in which one of the labeled-proteins is titrated against the other two components to generate a dose-response (DR) curve ... The salient characteristic of these binding interactions is a bell-shaped curve”. This description seems to be inaccurate as more typically the concentrations of each labelled target proteins would be kept constant and the PROTAC molecule would be titrated in dose-response, leading to a characteristic bell-shaped curve - in which, at low PROTAC concentrations insufficient bivalent exists to form the maximum achievable level of ternary complex, and at high concentrations binary interaction of PROTAC with either target protein begin to dominate, leading to a drop in ternary complex achieved. Titrating one labelled target protein to high concentration with a constant concentration of PROTAC would not typically cause such a hook effect, rather simply saturate at one end. Suggest amending this to refer to *titration of bivalent molecule/PROTAC*.

Once again, we thank reviewer for pointing this out. In the revised manuscript, we have changed the qualifying statement as follows:

“PROTAC-mediated ternary complex formation has been analyzed by proximity-based assays in which a bifunctional molecule is titrated against two labeled-proteins to generate a dose-response (DR) curve^{24, 26}. The salient characteristic of these binding interactions is a bell-shaped curve that describes the formation of a ternary complex and its dissolution due to competing binary interactions. Although these assays provide a means to evaluate the potency of PROTACs relative to one another, additional methods are required to characterize the thermodynamic and kinetic parameters of the complex²⁷.”

Line 93: “these methods preclude” – they do not preclude, but experiments can be designed in such a way as to avoid, the hook effect.

We thank reviewer for this suggestion. In the revised manuscript, we have modified the text as:

“These methods avoid measuring the competing binary interactions by using a preformed binary complex consisting of PROTAC and excess target protein (Figure 1C).”

Line 96: “ternary complex formation can be potently influenced by underlying protein-protein interactions (PPIs).” This is a valid statement, but in addition to protein-protein interactions also protein-ligand and -linker interactions for bivalent molecules have been shown to contribute to the interface of the complex (Eg. MZ1/pdb:5t35), it is suggested to broaden this statement to include these possibilities.

We agree with the reviewer that “in addition to protein-protein interactions also protein-ligand and -linker interactions for bivalent molecules have been shown to contribute to the interface of the complex”. We have modified the text in the revised manuscript as follows:

“SPR and ITC methods reveal that ternary complex formation can be potently influenced by both protein-protein interactions (PPIs) and protein-small molecule interactions.”

Line 161: The authors do not explain clearly why “In the SPR assay, $[T]_t \approx 25 \cdot K_{TP}$ for maintaining the binary complex”. Maintaining the target concentration at $\sim 25 \times K_D$ for the target/PROTAC interaction seems reasonable (albeit arbitrary), but the rationale for this is not clearly outlined in the text. Perhaps relating this to a concept such as fractional ligand occupancy may assist readers to implement this in their own work. In practice $\sim 25 \cdot K_{TP}$ has also not experimentally been used for all compounds evaluated in this study – eg. for compound 10 ($K_{TP} \sim 865$ nM) using a bromodomain target concentration of $5 \mu\text{M}$ actually represents $[T]_t \sim 6 \times K_{TP}$.

We thank reviewer for the careful reading of the manuscript, and the insightful comments which helped us to improve this manuscript. We have included the rationale in the *Appendix* for choosing $[T]_t \approx 25 \times K_{TP}$ in the SPR assay as follows:

Total target concentration is the sum of the free target $[T]$ and total bound target $[TP]$ as:

$$[T]_t = [T] + [TP] \quad (\text{A5})$$

Substituting eq A3 after rearrangement into eq A4 yields:

$$\frac{[TP]}{[P]_t} = \frac{[T]}{[T] + K_{TP}} \quad (\text{A6})$$

Eq A6 can be fitted to a hyperbolic binding curve (Figure A1) by plotting target concentrations as a multiplier of binary binding affinity (K_{TP}) against the fraction of binary complex. Figure A1

suggests that at $[T]_t \approx 25 \times K_{TP}$, saturation of PROTAC molecules ($\sim 96\%$) bound to the target $[TP]$ can be achieved. $[T]_t \cong [T]$, when total target concentration $[T]_t \gg [TP]$ to ensure the binary complex in SPR analysis.

Figure A1. Binding curve of varying concentrations of target (as a multiplier of binary binding affinity, K_{TP}) to fraction of binary complex.

Line 273: “Richling et al.” should be “Riching et al.”

We thank reviewer for spotting it. In the revised manuscript we fixed it.

“To build upon these SPR observations, a nanoBRET target engagement assay (Figure 6), similar to that published by Riching et al.,”

Line 412: “under saturating conditions” does not specify what is saturating

We thank reviewer for the suggestion. We modified the statement in the revised manuscript to make it clearer, as “under saturating conditions” was already discussed earlier in the text.

“The correlation between cooperativity and initial degradation rate for SMARCA2 degraders highlighted a difficult to predict but nonetheless critical attribute of our molecules.”

SI Lines 106 and 1109 “Availability Index” and “Permeability Index” – authors may wish to make these titles consistent

We once again thank reviewer for spotting it. We have modified the title of Table S9 and now it's consistent with Table S8.

“Table S8. SMARCA2 Degradable Availability Index”

“Table S9. BRD4 Degradable Availability Index”

SI Line 1032 “resulted into average diffraction with 3.7 Å resolution” – suggest “resulted in diffraction with average 3.7 Å resolution”

Once again, we thank reviewer for this suggestion. We have edited the modified Supplementary Information.

“Co-crystallization of compound 1 with SMARCA4/VBC resulted in diffraction with average 3.7 Å resolution.”

REVIEWERS' COMMENTS

Reviewer #5 (Remarks to the Author):

The authors have completed a comprehensive response to address the comments I had raised in my earlier review; I would support publication of the revised manuscript.